# BRAINCODEC: NEURAL FMRI CODEC FOR THE DECODING OF COGNITIVE BRAIN STATES

## ABSTRACT

Recently, leveraging big data in deep learning has led to significant performance improvements, as confirmed in applications like mental state decoding using fMRI data. However, fMRI datasets remain relatively small in scale, and the inherent issue of low signal-to-noise ratios (SNR) in fMRI data further exacerbates these challenges. To address this, we apply compression techniques as a preprocessing step for fMRI data. We propose BrainCodec, a novel fMRI codec inspired by the neural audio codec. We evaluated BrainCodec's compression capability in mental state decoding, demonstrating further improvements over previous methods. Furthermore, we analyzed the latent representations obtained through BrainCodec, elucidating the similarities and differences between task and resting state fMRI, highlighting the interpretability of BrainCodec. Additionally, we demonstrated that fMRI reconstructions using BrainCodec can enhance the visibility of brain activity by achieving higher SNR, suggesting its potential as a novel denoising method. Our study shows that BrainCodec not only enhances performance over previous methods but also offers new analytical possibilities for neuroscience. Our codes, dataset, and model weights are available at `https://anonymous.4open.science/r/BrainCodec`.

## 1 INTRODUCTION

In recent years, the scaling up of deep learning has significantly boosted performance in Natural Language Processing (NLP). Research on Scaling laws (Kaplan et al. (2020)) has shown that both parameters and data scale contribute to performance improvements. This finding has led to the release of numerous large-scale models and has influenced various research fields, including computer vision (Dosovitskiy et al. (2020); Radford et al. (2021)), speech processing (Wang et al. (2023); Le et al. (2024)), and neuroscience, which is the focus of this study.

Deep learning is actively employed in the field of neuroscience. For instance, numerous studies have proposed the use of deep learning in the analysis of fMRI data (Mozafari et al. (2020); Ozcelik et al. (2022); Takagi and Nishimoto (2023)). Recently, researchers have begun to publish their collected fMRI datasets, and efforts are underway to standardize the structure and preprocessing of neuroimaging data (Horien et al. (2021); Markiewicz et al. (2021); Gorgolewski et al. (2016); Esteban et al. (2019)). Against this backdrop, research on large-scale models using fMRI datasets has been increasing (Liu et al. (2023); Ortega Caro et al. (2023)).

Thomas et al. (2022) conducted the first study applying NLP modeling techniques to large-scale fMRI datasets. They utilized methods such as BERT (Devlin et al. (2018)) and Causal Sequence Modeling (CSM) (Brown et al. (2020)), proposing self-supervised learning (SSL) using large-scale raw fMRI data. By utilizing such pretrained models for fMRI feature extraction and applying them to downstream tasks, specifically for mental state decoding using the HCP dataset (Van Essen et al. (2013)), they demonstrated significant improvements in accuracy over baseline models. Studies of this nature, leveraging big data and SSL for fMRI feature extraction, have become increasingly prevalent in recent years (Asadi et al. (2023); Ortega Caro et al. (2023)).

Despite the advent of the big data era, the scale of fMRI data remains significantly smaller compared to fields such as NLP and computer vision, thus necessitating cleaner data. However, fMRI data are known for their low signal-to-noise ratio (SNR) due to the presence of multiple sources of noise (Geenjaar et al. (2022)). Thus, applying preprocessing that enhances the SNR of fMRI data will be crucial not only for these SSL methods but also for improving the performance of general analyses using fMRI in future research.

Here, we develop a codec model as a preprocessing method for fMRI data by importing the signal compression techniques from the field of audio, specifically known as the neural audio codec (NAC) (Kumar et al. (2024)). The NAC uses the Residual Vector Quantization (RVQ) method (Zeghidour et al. (2021)) to hierarchically discretize audio by tokenizing, achieving state-of-the-art performance in compression efficiency and quality (Wang et al. (2023); Song et al. (2024); Ju et al. (2024)).

In cognitive neuroscience, finding discrete representations is a natural approach because different functional signals are distributed across various brain areas or networks. For instance, DiFuMo (Dadi et al. (2020)) is a compression method that employs dictionary learning to obtain sparse representations. One of our motivations is to leverage the multi-scale discrete information inherent in BOLD signals. Multi-scale information is critical because it allows for capturing the hierarchical organization of brain functional signals, such as hierarchical properties of functional connectivities (Vidaurre et al. (2017)). RVQ is suited for this purpose. Furthermore, signal compression methods that incorporate spatiotemporal relationships can enhance the interpretability of noisy data (Descombes et al. (1998); Woolrich et al. (2004)). These findings encourage us to employ spatiotemporal compression techniques, such as NAC, to improve the SNR of fMRI data, leading to further performance enhancements.

Several data compression methods for fMRI data have been proposed. For instance, AutoEncoder-based compression methods using VAE or ViT (Dosovitskiy et al. (2020)) are also commonly seen (Chen et al. (2023); Kim et al. (2021)). While these methods can be used for the same purpose, they still have low compression rates due to continuous value-based compression methods, and they suffer from low interpretability.

**Our contributions are summarized as follows:**
• We propose **BrainCodec**, an RVQ-based codec model as a preprocessing method for fMRI data, inspired by the latest neural audio codec model.
• We evaluate the compression capability of BrainCodec as a preprocessing step within the framework of Thomas et al. (2022) and show that our method can further improve the classification performance of mental state decoding beyond conventional methods (Section 4.1).
• We analyze the codebooks, which are the latent representations obtained from the RVQ in BrainCodec. We propose obtaining these codebooks separately in two different domains: task fMRI and resting state fMRI, and comparing them. As a result, we elucidate their similarities and differences, confirming their validity from a neuroscientific perspective (Section 4.2.1). This result shows the high interpretability of BrainCodec.
• We analyze the performance changes when modifying the number of codebooks and find that competitive results are achieved even with half the codebooks, emphasizing the effectiveness of hierarchical modeling in RVQ. Additionally, we show that the reconstructions by BrainCodec provide clearer visualizations of brain activity, i.e., higher SNR, suggesting potential applications as a novel denoising technique (Section 4.2.2).
• We publish the codes, dataset, and model weights at `https://anonymous.4open.science/r/BrainCodec` to further development.

## 2 METHOD

### 2.1 BACKGROUND

Following Thomas et al. (2022), we consider CSM as a form of SSL for fMRI. CSM is a simple yet powerful method in NLP proposed by Radford et al. (2019). Let text symbols as $(s_1, s_2, \ldots, s_n)$. It is known that the

joint probability of this sequence can be decomposed as follows (Jelinek (1980)):

$$p(x) = \prod_{i=1}^{n} p(s_i | s_1, s_2, \ldots, s_{i-1}).$$

The authors proposed modeling this using only the Decoder of the Transformer model (Vaswani et al. (2017)). This approach allows for scaling up both the amount of data and the model size, achieving performance that was difficult with previous frameworks.

Thomas et al. (2022) proposed extending CSM to fMRI data, highlighting two main differences from CSM in NLP: (i) the direct use of raw fMRI data; and (ii) the introduction of TR embeddings.

**(i) Direct use of raw fMRI data**

fMRI data, which consist of four-dimensional BOLD signals over time, pose challenges in machine learning due to their high dimensionality (Lemm et al. (2011)). Brain activity exhibits strong spatial correlations (Smith and Kohn (2008)), allowing the aggregation of correlated voxels. The Dictionaries of Functional Modes (DiFuMo) (Dadi et al. (2020)) technique addresses this by defining a sparse dictionary matrix learned from millions of fMRI data. By using the pseudo-inverse of the DiFuMo matrix, the high-dimensional BOLD signals can be compressed into a lower-dimensional representation, allowing them to use fMRI data in NLP methods. Thus, they transform fMRI data into a form $X \in \mathbb{R}^{T \times n}$, where $n = 1024$, using the DiFuMo matrix, then pass it through a single Linear layer to serve as input to the CSM. They consider this the embedded representation of fMRI data, drawing an analogy to NLP. In this study, we also utilize DiFuMo.

**(ii) Introduction of TR embeddings**

While CSM employs positional embeddings to model temporal relationships, fMRI data may have varying repetition times (TRs), meaning the temporal relationship between each time-step is not constant. They propose explicitly including TR in the CSM input. Specifically, they prepare embeddings corresponding to $0, 0.2, 0.4, \ldots, 300s$. If the TR is $0.7s$ with a time length of 10, each time-step corresponds to $0, 0.7, 1.4, \ldots, 6.3s$. Each time-step is then matched with the nearest embedding, in this case, embeddings corresponding to $0, 0.6, 1.4, \ldots, 6.2s$. The resulting TR embeddings are added to the positional embeddings to model variations in TR. We also adopt this architecture and further introduce TR embeddings into our proposed method (See 2.2.1).

## 2.2 Proposed method

In this study, we propose BrainCodec, a codec model that considers spatiotemporal relationships (Fig 1). By compressing fMRI data using such a codec model, the training of models like CSM is facilitated, potentially improving performance. First, this section will detail BrainCodec. Then, it will describe how to use BrainCodec within the framework of SSL.

### 2.2.1 BrainCodec

BrainCodec features an autoencoder architecture consisting of an Encoder, a Residual Vector Quantization (RVQ) module (Zeghidour et al. (2021)), and a Decoder (Fig 1). First, the fMRI data transformed by DiFuMo is input into the Encoder, whose output (i.e. latent representation) is then discretized by the RVQ module. The discretized latent representation is subsequently fed into the Decoder, which outputs the fMRI data post-DiFuMo application. All modules are trained end-to-end to minimize reconstruction error. Below, we detail each module and the objective function.

**Encoder & Decoder modules.** The Encoder and Decoder are primarily composed of convolutional layers and are designed to be symmetrical. The Encoder takes the DiFuMo-transformed fMRI data $X \in \mathbb{R}^{T \times 1024}$ as input and produces a latent representation $Z \in \mathbb{R}^{3T \times 128}$. This involves spatial compression by a factor of $1024/128 = 8$ while expanding the temporal dimension by a factor of 3 through transposed convolution layers (Dumoulin and Visin (2016)). This results in an overall compression ratio of approximately 2.6,

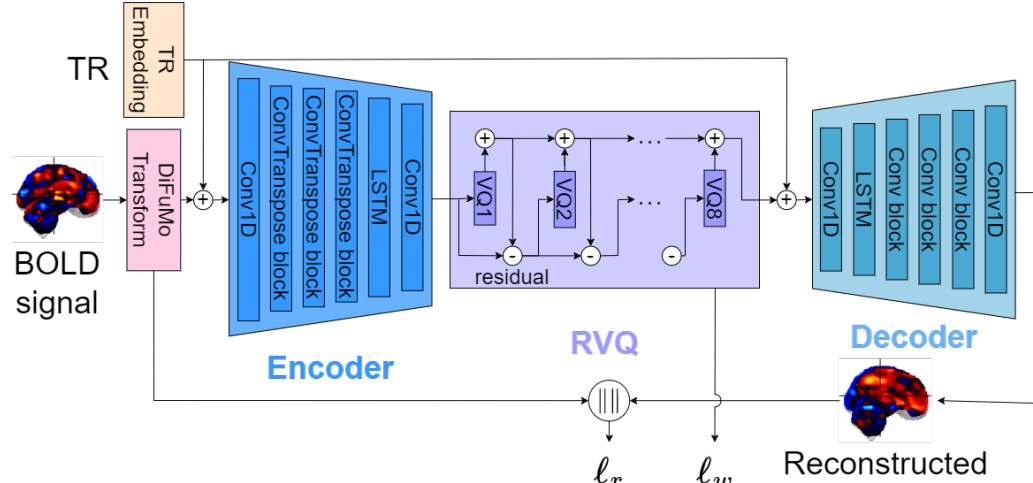

Figure 1: The architecture of BrainCodec: This model consists of an Encoder, a Decoder, and a Residual Vector Quantization (RVQ) module. The model input is data $X \in \mathbb{R}^{t \times 1024}$, transformed by DiFuMo from raw BOLD signals. It is trained through two losses: reconstruction error $\ell_r$ and commitment loss $\ell_w$ in RVQ.

slightly higher than Encodec's default compression ratio of $2.5 (= 320/128)$. For a detailed exposition of the architectural specifics, see Appendix A.1.

Additionally, we employ the TR embedding introduced in Section 2.1. Specifically, after creating the TR embedding, we add it to the inputs of both the Encoder and the Decoder. Since the input to the Decoder is temporally expanded, the TR embedding is also expanded accordingly.

**RVQ module.** Vector Quantization (VQ) is a discretization technique that finds the closest match to the input data from a codebook and returns its codebook IDs as the result. RVQ extends VQ by using multiple VQ in series, enabling efficient and finer discretization. Specifically, RVQ repeatedly discretizes the residual error left by the previous VQ (see Fig 1).

In this study, we use 8 codebooks. This means that the output of the Encoder, $Z \in \mathbb{R}^{3T \times 128}$, is converted into an ID sequence (token sequence) $Z_q \in \mathbb{R}^{3T \times 8}$. This approach achieves a much higher compression rate compared to continuous-value methods like VAE. During reconstruction, this token sequence is transformed back into embeddings using the codebook, and these vectors serve as the input to the Decoder.

**Training objective.** In this study, we utilize two training objectives: reconstruction error and RVQ commitment loss. For the reconstruction error, we use the L2 loss, which is the L2 distance between the DiFuMo-transformed fMRI data $X \in \mathbb{R}^{T \times 1024}$ and the Decoder output $\hat{X} \in \mathbb{R}^{T \times 1024}$. Additionally, the RVQ commitment loss is used to minimize the error during discretization. The final loss function $L$ that we employ is as follows:

$$\ell_r(x, \hat{x}) = \|x - \hat{x}\|_2, \quad \ell_w = \sum_{c=1}^{C} \|z_{c-1} - \text{RVQ}_c(z_{c-1})\|_2^2, \quad L = \ell_r(x, \hat{x}) + \ell_w,$$

where $x$ is the input fMRI data, $\hat{x}$ is the reconstructed fMRI data, $C$ is the number of codebooks of RVQ, $z_c$ is the $c$ th residual output in RVQ, $\text{RVQ}_c$ is the $c$ th codebook of RVQ.

It is important to note that we did not use a discriminator, which is commonly employed in conventional methods. Applying a discriminator to fMRI data resulted in unstable training due to the high level of noise inherent in the data. For more detailed experiments on the selection of these training objectives and other hyperparameters, please refer to Appendix A.2.

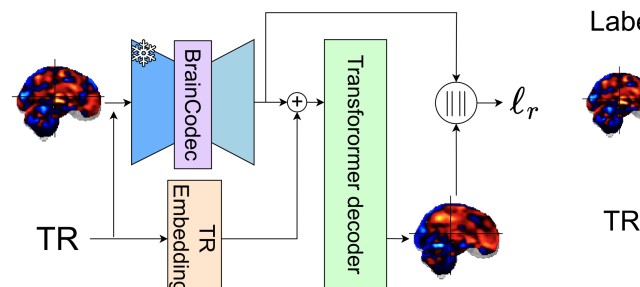

Figure 2: Upstream learning of CSM with Brain-Codec. BrainCodec is fixed. The training objective is L1 reconstruction loss ($\ell_r$).

Figure 3: Downstream learning of CSM with Brain-Codec. BrainCodec is also fixed. The training objective is cross-entropy loss ($\ell_c$).

### 2.2.2 CSM WITH BRAINCODEC

In this study, following the approach by Thomas et al. (2022), we train the CSM in two phases: (i) upstream learning and (ii) downstream learning. First, we perform self-supervised learning of CSM in the upstream learning. This enables the extraction of features from fMRI data. Then, using the model, we perform fine-tuning to solve the target classification task in the downstream learning.

**(i) Upstream learning.** The CSM undergoes self-supervised learning, meaning it is pretrained without specific labels. We reconstruct fMRI data using BrainCodec, and this reconstructed data is then used as both input and output for CSM (see Fig 2). The training objective is the reconstruction error, using the L1 loss. Except for processing fMRI data with BrainCodec, this method follows Thomas et al. (2022) without any modifications to the CSM architecture. This ensures a fair comparison with previous research.

**(ii) Downstream learning.** The pretrained CSM is fine-tuned for classification task. Specifically, we prepared a trainable class embedding (CLS) and concatenated it at the end of the input data (see Fig 3). This concatenated input is then processed by CSM, and the resultant CLS embedding is treated as a feature vector. This vector was input into a linear and softmax layer to produce outputs for the classification task.

## 3 EXPERIMENTAL SETUP

### 3.1 MODEL & HYPERPARAMETERS

**Architecture.** For details on BrainCodec, refer to Section 2.2.1. For more details, refer to Appendix A.1. In CSM, the dimensionality is 768, with 12 heads, 4 layers, and a dropout rate of 0.1. These parameters are identical to those used in the best model by Thomas et al. (2022).

**Training.** For BrainCodec training, we utilized the Adam optimizer with betas set to $(0.5, 0.9)$. The batch size was 64 and the learning rate was set at $1 \times 10^{-4}$. The fMRI data is randomly sampled to a length of 50 (with padding applied if necessary). This training was conducted on an Nvidia A100 with 40GB, requiring approximately one day. Note that all experiments are conducted using the Nvidia A100 with 40GB. For detailed training results, including the loss curve and reconstruction quality, please refer to Appendix A.3.

For CSM training, during upstream learning, we used the AdamW optimizer (betas=$(0.9, 0.95)$) with a batch size of 64 and a learning rate of $5 \times 10^{-4}$. A cosine scheduler was used, and training was taking about 14 hours. For downstream learning, we used different betas ($= (0.9, 0.999)$). This phase of training took approximately 3 hours. In both the upstream and downstream processes, similar to BrainCodec, the fMRI data is randomly sampled to a length of 50.

For more comprehensive training details and other configurations, refer to Appendix D and our publicly available GitHub repository.

## 3.2 DATASETS

The datasets utilized in this study differ between upstream and downstream learning. For upstream learning, 34 datasets were used, comprising 11,980 fMRI runs from 1,726 subjects, all of which are available on OpenNeuro.org (Markiewicz et al. (2021)). Specific identifiers for these data can be found in Appendix C.1. This dataset spans many acquisition sites and covers a diverse set of experimental conditions and domains. Notably, the data allocation for validation and testing was 4% and 1% respectively, selected randomly not from the total data but specifically within each dataset.

In downstream learning, three datasets were employed as benchmark mental state decoding datasets: the Human Connectome Project (HCP (Van Essen et al. (2013))), the Multi-Domain Task Battery (MDTB (King et al. (2019))) and Over 100 Task fMRI Dataset (Over100 (Tomoya Nakai (2020))). HCP includes task-fMRI data across 20 tasks from 100 subjects. The tasks include those related to working memory and emotion. MDTB contains data from 24 subjects who performed 26 tasks across two sessions, with each session including 8 fMRI runs. The tasks include concrete permuted rules operations (CPRO) and action observation. Over100 consists of 6 participants, each performing 103 cognitive tasks. Each participant completes 18 runs. The tasks include natural actions such as PressRight, where the participant presses a button with their right hand, and RestOpen, where they are asked to keep their eyes open. Additional details are available in Appendix C.2. The HCP and MDTB datasets were split by subject, with HCP divided into 48, 10, and 20 for train, validation, and test, respectively, and MDTB into 11, 3, and 9. In contrast, the Over100 dataset had only 6 subjects, so it was randomly split into 5,428, 904, and 1,810 runs.

It is important to note that both upstream and downstream datasets have been previously utilized and made publicly accessible by Thomas et al. (2022). These datasets were already preprocessed (details in Appendix C.4), and no further preprocessing was performed in our study. The data splitting ratios were also maintained as per the original study to ensure accurate comparisons.

## 4 RESULTS

### 4.1 DOWNSTREAM PERFORMANCE

In this section, we present the results of the downstream learning explained in Section 2.2.2. Specifically, we perform mental state decoding using datasets such as HCP, MDTB and Over100, comparing the performance on this classification task. It should be noted here that the upstream learning, which serves as a pretraining for downstream learning, has been confirmed to perform similarly to previous studies (see Appendix D).

#### 4.1.1 PREREQUISITES

We evaluated the performance of **Linear** model and **CSM**, following Thomas et al. (2022). The Linear model is a simple model that takes fMRI data, applies a linear layer to compress it in the time direction to 1, then applies another linear layer in the spatial direction before computing a softmax. It was considered state-of-the-art in the task of mental state decoding until the introduction of CSM (Schulz et al. (2020)). We also demonstrate the performance of this model when using BrainCodec. Specifically, instead of using raw fMRI data as input, we use the reconstructed fMRI data created by BrainCodec. Since the Linear model is very small, we did not perform upstream learning for it.

For comparison with BrainCodec, we have also prepared the following models:
**BrainCodec large:** In our preliminary research, it was found that increasing the number of codebooks in

Table 1: Decoding performances when combining Linear model & CSM with various codec models.

| Model | Codec | HCP | | MDTB | | Over100 | |
|-------|-------|-----|-----|------|------|---------|------|
| | | Acc | F1 | Acc | F1 | Acc | F1 |
| Linear | - | 0.622 | $0.616 \pm 0.161$ | 0.825 | $0.827 \pm 0.069$ | 0.259 | $0.156 \pm 0.138$ |
| Linear | BrainVAE | 0.734 | $0.721 \pm 0.165$ | 0.823 | $0.825 \pm 0.072$ | 0.282 | $0.177 \pm 0.148$ |
| Linear | BrainCodec large | 0.750 | $0.735 \pm 0.168$ | **0.828** | $\mathbf{0.829 \pm 0.069}$ | 0.317 | $0.206 \pm 0.154$ |
| Linear | BrainCodec | **0.814** | $\mathbf{0.784 \pm 0.133}$ | 0.811 | $0.812 \pm 0.072$ | **0.319** | $\mathbf{0.213 \pm 0.153}$ |
| CSM | - | 0.925 | $0.893 \pm 0.114$ | **0.901** | $\mathbf{0.897 \pm 0.054}$ | 0.413 | $0.318 \pm 0.169$ |
| CSM | BrainVAE | 0.924 | $0.892 \pm 0.131$ | 0.890 | $0.886 \pm 0.060$ | 0.408 | $0.308 \pm 0.171$ |
| CSM | BrainCodec large | 0.896 | $0.856 \pm 0.134$ | 0.898 | $0.895 \pm 0.058$ | 0.429 | $0.332 \pm 0.183$ |
| CSM | BrainCodec | **0.931** | $\mathbf{0.897 \pm 0.108}$ | 0.892 | $0.886 \pm 0.058$ | **0.518** | $\mathbf{0.437 \pm 0.172}$ |

the RVQ module improved the reconstruction error (Appendix A.2). Therefore, we prepared a version of BrainCodec with an extreme value of 128 codebooks.

**BrainVAE:** The VAE method is a commonly used compression technique in the field of neuroscience (Kim et al. (2021)). Here, it is employed to compare its continuous value-based approach with our discretization-based method. It removes the RVQ module, uses $\mu, \sigma$ as the Encoder's output, and $z = \sigma * \epsilon + \mu$ as the decoder's input, where $\epsilon \sim \mathcal{N}(0, I)$. The rest of the structure is the same.

Here, accuracy and F1 score (macro-averaged) are used as evaluation metrics. The accuracy is calculated as the average across all test data, while the F1 score is similarly computed over the entire test set, with the mean and standard deviation then calculated across different classes.

### 4.1.2 RESULTS WITH LINEAR MODEL

The Linear model was greatly affected by changes in weight decay due to its small parameters. The results shown here are with appropriately selected weight decay. For changes in performance due to weight decay variations, see Appendix E.

Table 1 demonstrates the performance variations of Linear models, when combined with different codec models. The proposed method, BrainCodec, shows notable improvements over the baseline and other methods for the HCP and Over100. On the other hand, BrainCodec, along with BrainVAE, underperforms compared to the baseline for the MDTB. These results indicate that no technique consistently performs well across all tasks, suggesting a significant dependency on the characteristics of the downstream dataset. Nevertheless, it is noteworthy that BrainCodec, when combined with the HCP and Over100, improved performance by approximately 31% and 23%, respectively, compared to the baseline. Since this method merely uses reconstructed data as input, it is easily integrated with existing techniques, making it a promising new preprocessing approach for combining with existing studies in neuroscience.

### 4.1.3 RESULTS WITH CSM

Table 1 shows that BrainCodec achieved improvements over the baseline in the HCP and Over100, underscoring its efficiency in compressing fMRI data while retaining essential information needed for accurate class classification. Notably, BrainCodec shows a significant performance improvement on the Over100, demonstrating its effectiveness in challenging tasks such as 100-class classification, where compression techniques prove particularly useful.

On the other hand, it is important to note that performance deteriorates not only with BrainCodec but also with all codec methods in MDTB. Compressing the input data using codec models results in the loss of detailed information, regardless of the method used. In MDTB, it is possible that such fine-grained information is

crucial. This indicates that codecs are not always effective, and the characteristics of the dataset must be considered.

In conclusion, BrainCodec was able to deliver significant performance improvements for datasets such as HCP and Over100. However, in neuroscience, the ability to analyze latent representations is equally important as improving performance in downstream tasks. In the following section, we will demonstrate that, as a result of incorporating the RVQ module, BrainCodec enables more flexible and effective analysis of latent representations compared to previous methods like VAE. In other words, BrainCodec is a promising method that combines both performance improvement and enhanced analytical capabilities.

## 4.2 CODEBOOK ANALYSIS

In this section, we analyze the codebooks obtained through BrainCodec training, discussing their latent representations and potential applications. Through these discussions, we aim to confirm the neuroscientific validity of the obtained codebooks and demonstrate their useful applications, which are challenging with conventional methods such as VAE. This highlights the novelty and importance of BrainCodec.

### 4.2.1 COMPARISON WITH RESTING STATE FMRI

We analyze the latent representations of the codebooks obtained through the training of BrainCodec. In the previous experiments, these codebooks have been trained using task fMRI data, which records brain activity during the performance of specific tasks. For comparative analysis, we also acquire codebooks using resting-state fMRI data, which represents brain activity in the absence of explicit tasks. By comparing codebooks learned from two different domains in this manner, we can interpret the latent representation. We will quantitatively and qualitatively evaluate the differences and similarities between these codebooks and discuss their validity from a neuroscientific perspective.

We collected resting state datasets from Open-Neuro.org and the SRPBS Traveling Subject MRI Dataset (Tanaka et al. (2021)), aggregating a total of 9 datasets comprising 1230 fMRI data points from 389 subjects. Details of these datasets are provided in Appendix C.3. Using the same training protocols and parameters, we trained BrainCodec on this dataset. Subsequently, we compared the codebooks from BrainCodec trained on resting state fMRI data (referred to as Rest-codebook) with those from BrainCodec trained on task fMRI data (Task-codebook), which is used in Section 4.1. As a reference, we also generated Gaussian noise sampled from a standard normal distribution, matching the shape of the codebooks (Random-codebook).

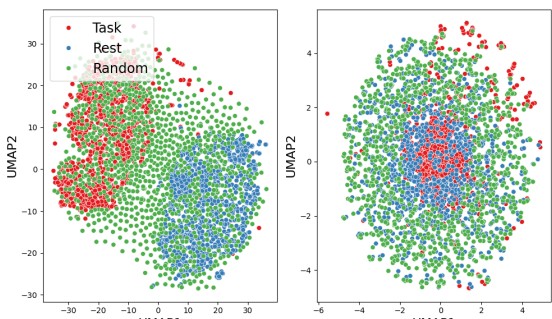

Figure 4: The UMAP visualization compares Task-codebook (red), Rest-codebook (blue), and Random-codebook (green). The left plot shows the first codebook, while the right plot shows the second codebook.

First, we visualized the codebook in Figure 4, using the dimensionality reduction method UMAP (McInnes et al. (2018)). The left plot corresponds to the first codebook, while the right plot corresponds to the second one. The subsequent codebooks are nearly identical to the second plot. For all results, refer to Appendix F.1.

In the first codebook, it can be observed that the Task-codebook and Rest-codebook form distinct clusters. In recent years, there has been a growing discussion that analyzing brain activity specific to task fMRI is more beneficial than using resting-state fMRI (Greene et al. (2018); Finn (2021)). Therefore, by utilizing only the first codebook, BrainCodec has the potential to isolate precisely those distinctive aspects of task fMRI. On the

other hand, the subsequent codebooks exhibit an inclusive relationship, where task-induced brain activities fall within the range of resting-state activities, consistent with Luczak et al. (2009).

To examine this second point from a quantitative perspective, we compare the downstream performance. In this case, we use the Linear+BrainCodec trained by HCP as the baseline, but during inference, we replace the codebooks with Rest-codebook.

Table 2: The change in accuracy when modifying the codebook during inference.

| codebook | Acc |
|---|---|
| Task-codebook | 0.814 |
| Rest-codebook | **0.614** |
| w/o codec | 0.187 |

The results are shown in Table 2. As can be observed, the decrease in accuracy when switching to the Rest-codebook is significantly smaller compared to the case where no codec is used. Despite the differences in both the domain and the volume of the datasets, the small decrease in accuracy suggests a similarity between the two codebooks, quantitatively verifying the claims made by Luczak et al. (2009). As another quantitative measure, we calculated Sinkhorn distance between each codebooks (Appendix F.2).

### 4.2.2 Performance Comparison and Reconstruction Using Partial Codebooks

In the Linear+Encodec, we investigate how changing the number of codebooks used for data reconstruction affects downstream performance. The results are shown in Table 3. As can be seen, initially, increasing the number of codebooks enhances performance, with the best performance occurring when all codebooks are used. This indicates that the codebook obtained through BrainCodec contains no superfluous information and that useful information has been learned up to the topmost codebook. Amazingly, using half the layers achieves performance nearly equivalent to using all layers, and even using just one layer surpasses the baseline performance. This is because, according to the principles of RVQ, the lower layers tend to capture the primary variations (Zhang et al. (2023)), indicating that sufficient information for class classification is concentrated in the lower half.

Table 3: In the Linear + Brain-Codec method, the performance change of the downstream task (using HCP) when using different codebooks.

| codebook number | Acc |
|---|---|
| - (baseline) | 0.622 |
| 0 | 0.632 |
| 0,1,2,3 | 0.798 |
| 0,1,2,3,4,5,6,7 | **0.814** |

Subsequently, we analyzed reconstructed fMRI data using a part of the codebooks (Figure 5). Here, we focus on the MOTOR task in the HCP. In the MOTOR task, participants are visually instructed to tap their fingers, curl their toes, or move their tongue. Here, we show the fMRI data corresponding to the tongue instruction.

Compared to the original fMRI data, the reconstructed data using BrainCodec, regardless of the number of codebooks, clearly shows an initial response in the visual cortex induced by the visual instruction. Following this, the motor cortex also appears to respond more distinctly than in the original. Additionally, noisy responses seen in the original are absent in the reconstructed data. This trend was also observed in tasks other than motor tasks (Appendix F.3).

In contrast, BrainVAE produces reconstructions that are very close to the original, making qualitative interpretation challenging. Furthermore, using only half of the codebooks yields results similar to those obtained using all codebooks, which is consistent with downstream performance. Using fewer codebooks achieves higher compression, potentially resulting in a better signal-to-noise ratio.

Table 4: The L1 distances for Mean and each method.

| method | L1 dist. |
|---|---|
| original | $0.751 \pm 0.042$ |
| BrainVAE | $0.798 \pm 0.032$ |
| BrainCodec (0-7) | $0.363 \pm 0.039$ |
| BrainCodec (0-3) | **$0.349 \pm 0.032$** |

Visualizing task-related responses generally requires averaging many fMRI data sets (Barch et al. (2013)). In fact, when comparing the original fMRI data from a single run (Fig. 5, far left) with the fMRI data averaged over 78 runs of

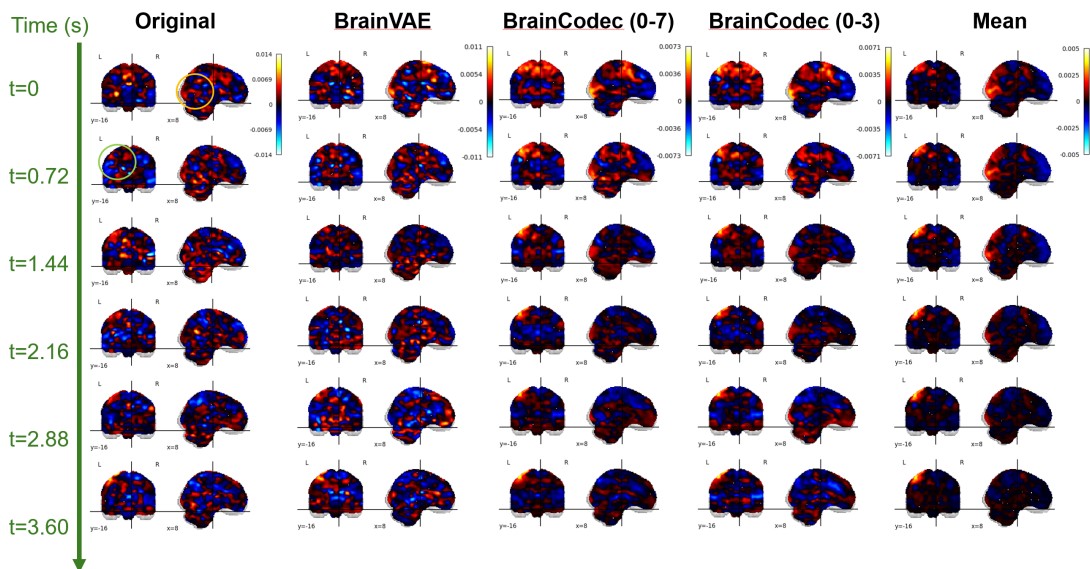

Figure 5: Comparison of the original fMRI from a MOTOR task trial (tr=0.72) in the HCP dataset with its reconstructed fMRI using BrainVAE, BrainCodec, and the averaged fMRI over 78 runs (Mean). BrainCodec uses the full codebooks (0-7), and the first half (0-3). The regions encircled by the orange indicate the visual cortex, while the green denote the tongue motor area.

this condition (Mean, Fig. 5, far right), it is evident that brain activity is more clearly visible in the Mean. On the other hand, BrainCodec can produce comparable results with just a single specific run.

This is evaluated using objective metrics. Table 4 shows the L1 distances between Mean and each method. This result validate that BrainCodec achieves a high SNR. Thus, this approach has the potential to become a more convenient and high-performance new denoising method. Of course, there are still some unclear fMRI data even after processing through BrainCodec, and there is no guarantee that they always indicate the correct responses. Further verification and improvements remain tasks for the future.

## 5 CONCLUSION

In this study, we proposed a compression method for fMRI, BrainCodec. As a result, we demonstrated that combining BrainCodec with SOTA models such as Linear and CSM can achieve further improvements in accuracy of mental state decoding. Additionally, we quantitatively and qualitatively compared the codebooks obtained from task fMRI and resting state fMRI, elucidating common and different aspects in representations for each codebook. We confirmed that these aspects are consistent with neuroscientific findings. Also, we focused on the performance changes resulting from varying the number of codebooks used. We demonstrated that using only half of the layers achieves performance comparable to prior research, highlighting the high compression capability of BrainCodec. Furthermore, comparisons of fMRI reconstructions with other methods showed that BrainCodec can clearly extract brain activity, suggesting its potential as a novel denoising method.

Through this study, we believe the necessity for large-scale fMRI data will only increase. One major obstacle in collecting such data is the issue of privacy. However, by using this compression method to exchange data in a compressed state, we may see a potential solution to this problem. Therefore, this paper could contribute not only to neuroscience but also to the field of deep learning itself.

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

# Appendices

## A    BRAINCODEC DETAILS

### A.1    MODEL ARCHITECTURE

Table 5: Hyper-parameters of BrainCodec

| Module | Hyper-Parameter | Value |
|---|---|---|
| Encoder | ConvTransposedBlock Number | 3 |
| | ConvTransposedBlock Filter Sizes | [1024, 512, 256] |
| | ConvTransposedBlock Strides | [1, 1, 3] |
| | Activate Function | ELU (Clevert et al. (2015)) |
| | LSTM Number | 2 |
| Residual Vector Quantizer | Codebook Number | 8 |
| | Codebook Dim | 128 |
| | Codebook Vocabulary | 1024 |
| Decoder | ConvBlock Number | 3 |
| | ConvBlock Filter Sizes | [256, 512, 1024] |
| | ConvBlock Strides | [3, 1, 1] |
| | Activate Function | ELU (Clevert et al. (2015)) |
| | LSTM Number | 2 |

In this section, we delineate the architecture of the model, focusing on the parameters without delving into the specifics of parameter selection. For further details on parameter choices, refer to A.2.

The critical hyperparameters are summarized in Table 5. As illustrated in Figure 1, BrainCodec is an AutoEncoder-based model consisting of three primary components: the Encoder, the Decoder, and the Residual Vector Quantization (RVQ).

The Encoder is primarily composed of ConvTransposedBlocks. Each ConvTransposedBlock consists of a Transposed Convolution Layer (Dumoulin and Visin (2016)) and a Residual Layer. In the Transposed Convolution Layer, operations are performed to expand in the temporal direction when the stride is greater than one. Additionally, a standard Convolution Layer is inserted to match the input and output dimensions, and an LSTM is applied to the output of the ConvTransposedBlock. Summarizing the transformations, the shape of the input fMRI data after applying DiFuMo (1024, 50) undergoes the following changes:

$$
\begin{aligned}
(1024, 50) \quad &\rightarrow \text{Convolution layer} \quad &&\rightarrow (1024, 50) \\
\rightarrow (1024, 50) \quad &\rightarrow \text{ConvTransposedBlock1} \quad &&\rightarrow (1024, 50) \\
\rightarrow (1024, 50) \quad &\rightarrow \text{ConvTransposedBlock2} \quad &&\rightarrow (512, 50) \\
\rightarrow (512, 50) \quad &\rightarrow \text{ConvTransposedBlock3} \quad &&\rightarrow (256, 150) \\
\rightarrow (256, 150) \quad &\rightarrow \text{LSTM} \quad &&\rightarrow (256, 150) \\
\rightarrow (256, 150) \quad &\rightarrow \text{Convolution layer} \quad &&\rightarrow (128, 150)
\end{aligned}
$$

The Decoder, on the other hand, mirrors the Encoder's structure. Each module is designed symmetrically with reversed filter sizes and strides, and the LSTM is applied first. Unlike the Encoder, the Decoder compresses in the temporal direction using standard Convolution Layers instead of Transposed Convolution Layers.

Finally, the RVQ is an extension of the widely used Vector Quantization (VQ) module, concatenating multiple VQ modules (eight in this case). Each VQ's input is the difference between the input and output of the previous VQ. This hierarchical modeling allows different codebooks to capture distinct features, akin to frequency decomposition. In the audio domain, the first VQ module tends to capture low-frequency semantic features, enabling techniques like teacher forcing to explicitly learn these semantic features, as proposed in codecs such as the SpeechTokenizer (Zhang et al. (2023)). Similar trends are observed in our study, as demonstrated in Section 4.2.

## A.2 Hyper-Parameter selections

In this section, we present the important hyperparameters considered in our study, accompanied by loss graph plots. It is important to note that while each individual experiment offers a valid comparison, the overall comparisons are not consistent across all experiments. Therefore, direct comparisons between separate experiments should be approached with caution. Additionally, since the trajectory of the loss typically stabilizes early in the training process, some experiments were halted at the initial stages.

**Figure 6a: Strides**
Our experiments revealed that increasing the temporal dimension in the encoder leads to a greater reduction in reconstruction error. This is a logical outcome, as it essentially corresponds to a reduction in effective compression rate, making reconstruction easier. Considering this trade-off, we opted for a factor of 3 for temporal expansion. This choice is analogous to the approach used in Encodec, where the temporal dimension is compressed by a factor of 320, while the spatial dimension is compressed by a factor of 128, resulting in an effective compression rate of approximately 2.5. In BrainCodec, we used the spatial dimension compression to 128, as in Encodec, yielding an 8-fold compression. To achieve an effective compression rate closest to 2.5, we chose to expand the temporal dimension by a factor of 3.

**Figure 6b: Loss function**
By default, Encodec employs L1 loss, which is a very common setting in the audio domain. However, this setting is not self-evident for fMRI data. In fact, in reconstructions using VAE, L2 loss is also often employed (Kim et al. (2021)). Therefore, we conducted a comparison and found that using L2 loss resulted in a lower reconstruction error for L2 loss, while L1 loss increased correspondingly, and vice versa. This outcome is quite natural. Given that it is unclear which loss function is better based on reconstruction error alone, we compared them in terms of downstream performance. As shown in the figure, L2 loss contributed slightly more to improvement. L2 loss imposes a heavier penalty on outliers, which could be beneficial for the noisy nature of fMRI data that may contain many outliers.

**Figure 6c: Discriminator**
This is likely the most significant factor contributing to the differences observed. We concluded not to use the traditionally well-utilized discriminator. The reason, as illustrated in the figure, is that it exacerbated the reconstruction error. This issue was not resolved by simply adjusting the coefficients between the loss functions. This instability is believed to stem from the inherently noisy nature of fMRI data. Reconstruction error, by definition, calculates loss based on the difference across all data points between input and output. Consequently, when inputs like fMRI data, which are significantly affected by noise, are processed, the learning progresses towards replicating that noise in the output. While discriminators generally aim to enhance fidelity, employing them leads to learning that replicates a similar noise output from noise inputs, essentially attempting to reproduce the noise distribution. Therefore, this compatibility issue between the reconstruction error, which aims to fit to one realization of noise distribution, and the use of discriminators led to unstable learning outcomes. To improve upon this, approaches such as the introduction of noise-robust GANs (Kaneko and Harada (2020)) might be considered, though this remains a topic for future research.

Additionally, as shown in Figure 6d, changing the number of ConvBlocks confirms that with fewer layers, overfitting occurs rapidly. Furthermore, altering the dimensions of the codebook (Figure 6e) did not result

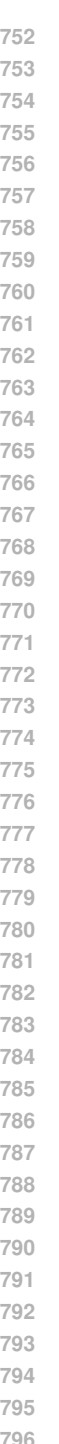

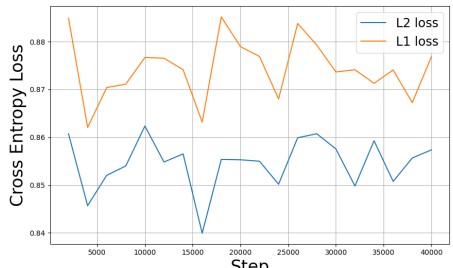

(a) Changes in reconstruction error when the stride of ConvBlock is altered.

(b) Changes in cross-entropy loss (valid) on the HCP dataset when the loss function is altered.

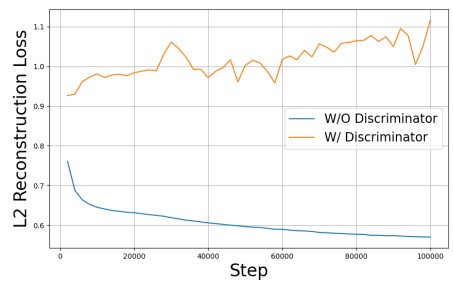

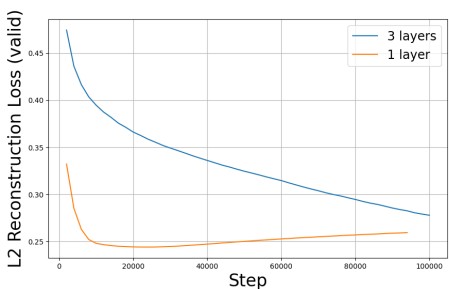

(c) Changes in reconstruction error when using and not using a discriminator.

(d) Changes in reconstruction error when the number of ConvBlocks is altered.

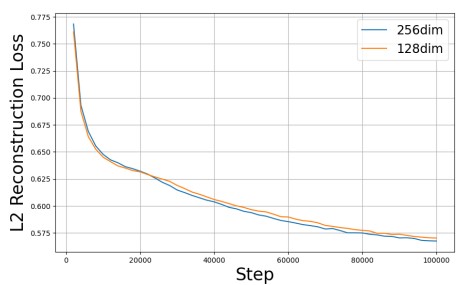

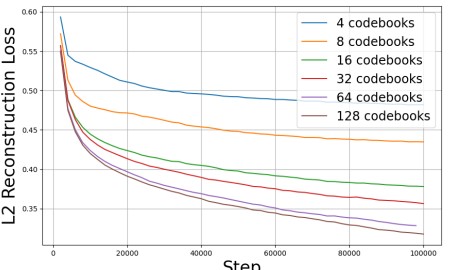

(e) Changes in reconstruction error when the dimensions of the codebook are altered.

(f) Changes in reconstruction error when the number of codebooks is altered.

Figure 6: Changes in the loss curve when varying hyperparameters.

in significant differences, so we used the default 128 dimensions. Lastly, we also present the results of changing the number of codebooks (Figure 6f). Naturally, the reconstruction error decreases as the number of codebooks increases. However, increasing the number of codebooks reduces the difference from continuous value compression and makes the analysis significantly more complex. Therefore, we adopted the default of 8 codebooks for our proposed method.

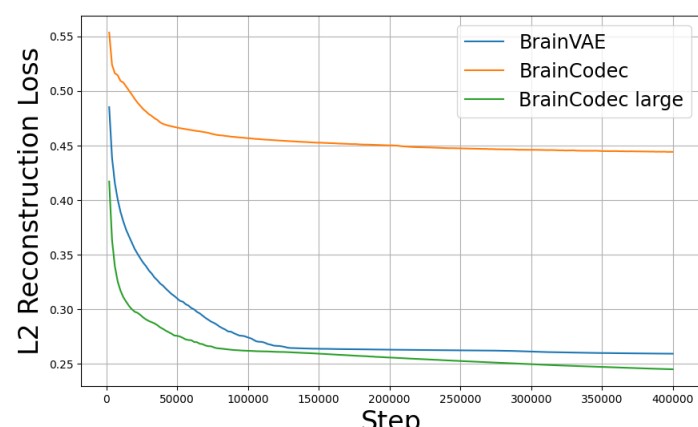

Figure 7: We present the reconstruction errors (L2 loss) during validation for BrainCodec, BrainCodec large, and BrainVAE.

### A.3 TRAINING RESULTS OF BRAINCODEC

Here, we present the reconstruction errors and actual examples of reconstructions for BrainCodec, BrainCodec large, and BrainVAE, which were used in Section 4.1.

Figure 7 shows the reconstruction errors. BrainVAE, which uses continuous latent representations, and BrainCodec large, which uses a very high number of codebooks (128), have significantly lower reconstruction errors compared to BrainCodec. However, as discussed in Section 4.1, there are still cases where improvements depend on the task. This indicates that further improvements are possible, though it is unclear whether this should focus on further reducing reconstruction errors or introducing discriminators. This remains a topic for future research.

Figure 8 displays the fMRI images reconstructed by each method alongside the ground truth images. The images reconstructed by BrainCodec are noticeably rough and fail to capture finer details. In contrast, the reconstructions by BrainCodec large and BrainVAE, as reflected in their lower loss, manage to preserve finer details more accurately.

While these qualitative results are not conclusive, it is noteworthy that discretizing with only 8 codebooks, as in BrainCodec, leads to rougher reconstructions. This suggests that using a higher number of codebooks, as in BrainCodec large, or continuous latent representations, as in BrainVAE, may be more effective for capturing detailed brain activity in fMRI data.

## B TOKENCSM

### B.1 ABOUT TOKENCSM

By using BrainCodec, we can convert fMRI data into a token sequence. This approach allows us to treat fMRI data similarly to text information, enabling the full utilization of NLP frameworks. Specifically, we perform next token prediction using the obtained token sequence as input and train the model with cross-entropy loss (see Fig 9). We refer to the CSM trained in this manner as **TokenCSM**. This method is also commonly used in the field of audio area (Wang et al. (2023); Copet et al. (2024)).

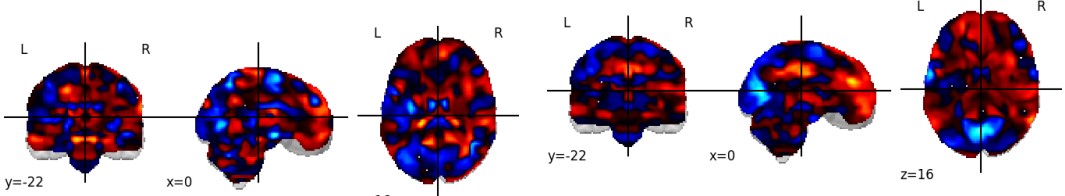

(a) The image of the ground truth fMRI data.

(b) The image of the reconstructed fMRI data by Brain-Codec.

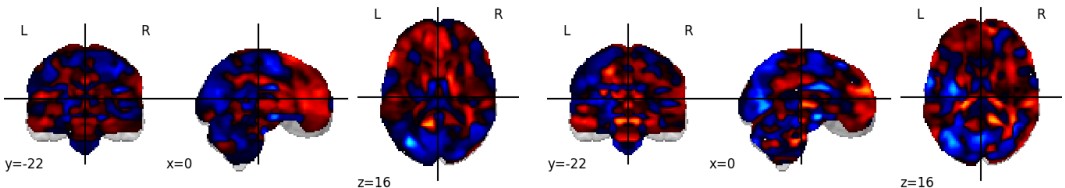

(c) The image of the reconstructed fMRI data by Brain-Codec large.

(d) The image of the reconstructed fMRI data by Brain-VAE.

Figure 8: The images of ground truth and reconstructed fMRI data.

## B.2 MODEL ARCHITECTURE & TRAINING METHOD

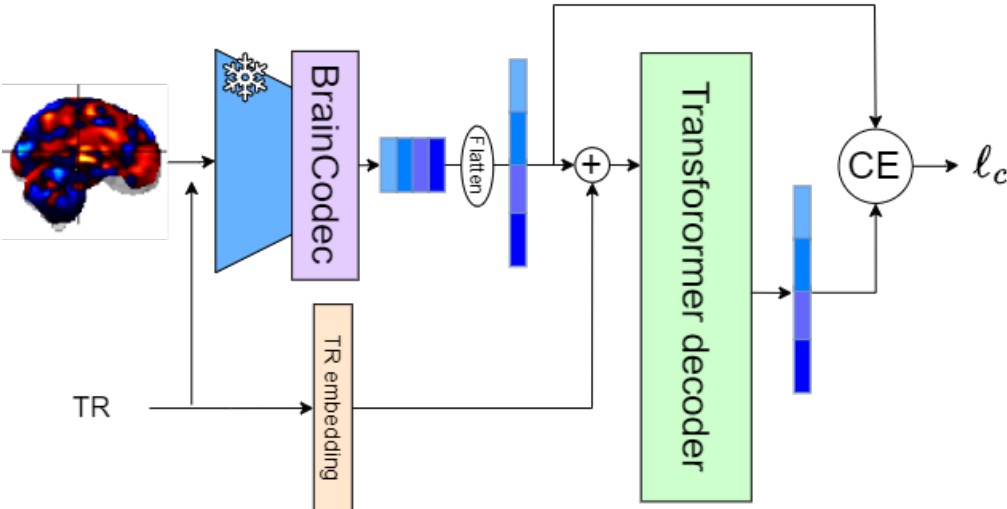

Figure 9: Upstream learning of TokenCSM.

Here, we describe how to train CSM in a token-based manner by leveraging BrainCodec's ability to tokenize fMRI data. This technique is well-known for handling raw data that is difficult to work with directly, such as audio (Wang et al. (2023); Łajszczak et al. (2024)).

Using the Encoder and RVQ of BrainCodec, DiFuMo-transformed fMRI data with a shape of (1024, 50) can be converted into a token sequence with a shape of (8, 150). Typically, in Transformer training with text, the token sequence shape is a one-dimensional array like (150,). The hierarchical tokenization introduced by RVQ must be managed appropriately. MusicGen (Copet et al. (2024)) discusses how to input these hierarchical token sequences into a Transformer. It shows that the "delay pattern", which inputs the token sequences offset by one token per hierarchy, best balances efficiency and performance. However, our experiments found that the "flatten pattern", which simply arranges the token sequences in a single line, yielded the best results (see Fig 10b).

Figure 9 illustrates the overall process of TokenCSM's upstream learning. Note that, since the input and output are token sequences, the training objective is cross-entropy loss rather than reconstruction error. For downstream learning, as described in Section 2.2.2, we append a CLS token at the end and use this CLS token as the feature for solving the classification task.

For TokenCSM training, the parameters for upstream learning were the same as CSM (see Section 3.1), except the batch size was 32 and the learning rate was $7 \times 10^{-4}$. This training lasted about 56 hours. For downstream learning, the only change made compared to the first method was the batch size, which was decreased to 64. This phase of training took approximately 8 hours. This configuration results in approximately 100 million parameters. Compared to the tens of billions of parameters commonly used in NLP (Touvron et al. (2023)) and the hundreds of millions to billions used in speech models (Łajszczak et al. (2024)), this is still small.

### B.3 HYPER-PARAMETER SELECTIONS

Here, we introduce the loss curves for TokenCSM when hyperparameters are adjusted. Specifically, we illustrate the effects of varying the number of layers and token rearrangement patterns.

Figure 10a shows the loss curves when the number of layers is varied. It was observed that increasing the number of layers consistently reduces the loss. Naturally, increasing the number of layers, and hence the parameter size, necessitates a corresponding increase in data volume. Since the data volume was fixed in this study, we selected 24 layers, the maximum value shown here.

Figure 10b depicts the changes in loss when the token rearrangement patterns "delay" and "flatten" are applied. As shown, the "flatten" pattern significantly outperforms the "delay" pattern. However, it is important to note that the "flatten" pattern uses approximately 8 times more token length than "delay," resulting in increased computational resources and time.

Moreover, regardless of the method, the absolute loss value remains around 6. Initially, the loss was approximately 6.8, so this represents only a reduction of 0.8. In text-to-speech systems, which operate on similar principles, the cross-entropy loss typically ranges from 3 to 4. This indicates that the current method has considerable room for improvement. The most effective and reliable way to achieve this improvement is to increase both the data volume and parameter size. Collecting more data remains a significant future challenge for this field.

### B.4 DOWNSTREAM PERFORMANCE

The downstream training for TokenCSM is conducted in exactly the same manner as the training method described for CSM in Section 2.2.2.

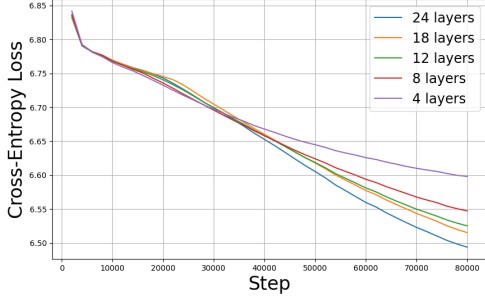
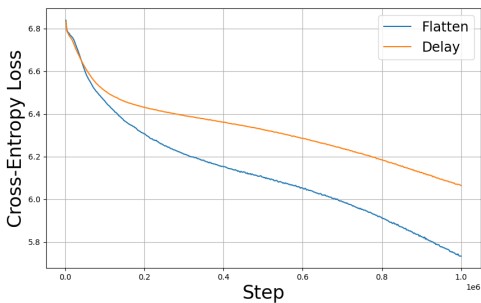

(a) The variation in cross-entropy loss for TokenCSM as the number of layers is adjusted.

(b) The variation in cross-entropy loss for TokenCSM as the token pattern is changed.

Figure 10: Loss curves of the upstream learning of TokenCSM.

Table 6: Decoding performances of TokenCSM.

| Model | HCP | | MDTB | |
|---|---|---|---|---|
| | Acc | F1 | Acc | F1 |
| CSM | 0.925 | $0.893 \pm 0.114$ | 0.901 | $0.897 \pm 0.054$ |
| TokenCSM | 0.496 | $0.282 \pm 0.268$ | 0.700 | $0.683 \pm 0.127$ |

Table 6 shows the downstream performance of TokenCSM on the HCP and MDTB datasets. the performance significantly deteriorated from the baseline in both the HCP and MDTB datasets under the current settings. This is believed to be due to the sheer volume of data and the size of the model. While the data used in this study was limited to about 10,000, with a parameter size of 100M, the initial CSM-based method, GPT-2 (Radford et al. (2019)), had a parameter size of 1.5B and utilized 40GB of data from vast amounts of webpages. If the goal is solely to improve downstream task performance, such large models and datasets might not be necessary. However, the absolute values of the loss in upstream learning for TokenCSM were not only large but also showed minimal reduction (Appendix B.3). Expanding the fMRI dataset size by tens of times in the future and applying such large-scale methods will be an important direction for further improving accuracy.

## C  DATASET DETAILS

### C.1  DATASETS FOR UPSTREAM TASK

Table 7 presents a summary of the datasets included in our upstream training. This information is cited from Appendix Table 1 in Thomas et al. (2022). The unpreprocessed fMRI data for all datasets are publicly accessible under a Creative Commons CC0 license via OpenNeuro.org (Markiewicz et al. (2021)), with each dataset identified by a specified identifier (ID). All fMRI data were de-identified and collected and shared with human consent in a manner approved by institutional review boards. No personally identifiable data were used. The preprocessed datasets and the code to utilize them in PyTorch can be found in the GitHub repository published by Thomas et al. (2022): `https://github.com/athms/learning-from-brains`.

Table 7: Overview of upstream datasets. For each dataset, the OpenNeuro.org identifier and DOI are provided, along with the number of individuals and fMRI runs included in our upstream dataset, a brief text descriptor, and the DOI of an associated publication.

| ID | DOI | #Individuals | #Runs | Text descriptor | DOI publication |
|---|---|---|---|---|---|
| ds000003 | 10.18112/openneuro.ds000003.v1.0.0 | 13 | 13 | Rhyme judgment | 10.1162/jocn.2007.19.10.1643 |
| ds000009 | 10.18112/openneuro.ds000009.v1.0.0 | 24 | 144 | The generality of self-control | unpublished |
| ds000030 | 10.18112/openneuro.ds000030.v1.0.0 | 144 | 1029 | UCLA Consortium for Neuropsychiatric Phenomics LA5c Study | 10.1038/sdata.2016.110 |
| ds000113 | 10.18112/openneuro.ds000113.v1.3.0 | 20 | 976 | Study Forrest | 10.1038/sdata.2014.3 |
| ds000140 | 10.18112/openneuro.ds000140.v1.0.0 | 33 | 297 | Distinct brain systems mediate the effects of nociceptive input and self-regulation on pain | 10.1371/journal.pbio.1002036 |
| ds000157 | 10.18112/openneuro.ds000157.v1.0.0 | 28 | 28 | Block design food and non-food picture viewing task | 10.1016/j.bbr.2013.03.041 |
| ds000212 | 10.18112/openneuro.ds000212.v1.0.0 | 39 | 370 | Moral judgments of intentional and accidental moral violations across Harm and Purity domains | 10.1073/pnas.1207992110 |
| ds000224 | 10.18112/openneuro.ds000224.v1.0.3 | 10 | 767 | The Midnight Scan Club (MSC) dataset | 10.1016/j.neuron.2017.07.011 |
| ds001132 | 10.18112/openneuro.ds001132.v1.0.0 | 45 | 15 | Watching BBC's Sherlock | 10.1038/nn.4450 |
| ds001145 | 10.18112/openneuro.ds001145.v1.0.0 | 24 | 24 | Watching The Twilight Zone | 10.1093/cercor/bhv155 |
| ds001499 | 10.18112/openneuro.ds001499.v1.3.1 | 4 | 515 | BOLD5000 | 10.1038/s41597-019-0052-3 |
| ds001612 | 10.18112/openneuro.ds001612.v1.0.2 | 23 | 135 | Offline replay supports planning in human reinforcement learning | 10.7554/eLife.32548 |
| ds001715 | 10.18112/openneuro.ds001715.v1.0.0 | 34 | 407 | Dissociable neural mechanisms track evidence accumulation for selection of attention versus action | 10.1038/s41467-018-04841-1 |
| ds001734 | 10.18112/openneuro.ds001734.v1.0.5 | 108 | 431 | Neuroimaging Analysis Replication and Prediction Study (NARPS) | 10.1038/s41597-019-0113-7 |
| ds001882 | 10.18112/openneuro.ds001882.v1.0.0 | 19 | 150 | Social Decision-Making Intertemporal Choice Task Dataset | 10.7554/eLife.44939 |
| ds001883 | 10.18112/openneuro.ds001883.v1.0.3 | 20 | 158 | Social Decision-Making Risky Choice Task Dataset | 10.7554/eLife.44939 |
| ds001921 | 10.18112/openneuro.ds001921.v1.0.0 | 15 | 30 | Anterior cingulate engagement in a foraging context reflects choice difficulty, not foraging value (1) | 10.1038/nn.3771 |
| ds001923 | 10.18112/openneuro.ds001923.v1.0.0 | 14 | 42 | Anterior cingulate engagement in a foraging context reflects choice difficulty, not foraging value (2) | 10.1038/nn.3771 |
| ds002306 | 10.18112/openneuro.ds002306.v1.0.3 | 6 | 102 | Over 100 Task fMRI Dataset | 10.1038/s41467-020-14913-w |
| ds002345 | 10.18112/openneuro.ds002345.v1.1.4 | 343 | 861 | Narratives Collection | 10.1038/s41597-021-01033-3 |
| ds002685 | 10.18112/openneuro.ds002685.v1.3.1 | 11 | 1263 | Individual Brain Charting | 10.1038/sdata.2018.105 |
| ds002785 | 10.18112/openneuro.ds002785.v2.0.0 | 216 | 1235 | Amsterdam Open MRI Collection-PIOP1 | 10.1038/s41597-021-00870-6 |
| ds002790 | 10.18112/openneuro.ds002790.v2.0.0 | 225 | 887 | Amsterdam Open MRI Collection-PIOP2 | 10.1038/s41597-021-00870-6 |
| ds002841 | 10.18112/openneuro.ds002841.v1.0.1 | 29 | 169 | Intuitive physics with fMRI | 10.7554/eLife.46619 |
| ds002995 | 10.18112/openneuro.ds002995.v1.0.1 | 18 | 192 | Taste Quality Representation in the Human Brain | 10.1523/JNEUROSCI.1751-19.2019 |
| ds003085 | 10.18112/openneuro.ds003085.v1.0.0 | 39 | 156 | Temporal Dynamics of Emotional Music | 10.1016/j.neuroimage.2019.116512 |
| ds003089 | 10.18112/openneuro.ds003089.v1.0.1 | 20 | 40 | Somatosensory phase-encoded bilateral full-body light touch stimulation | 10.1016/j.neuroimage.2020.117257 |
| ds003148 | 10.18112/openneuro.ds003148.v1.0.1 | 35 | 412 | Neuroimaging evidence for network sampling Theory of Human Intelligence | 10.1038/s41467-021-22199-9 |
| ds003242 | 10.18112/openneuro.ds003242.v1.0.0 | 95 | 598 | MRI data of 40 adult participants in response to a cue-induced craving task following food fasting, social isolation, and baseline (within-subject design) | 10.1038/s41593-020-00742-z |
| ds003338 | 10.18112/openneuro.ds003338.v1.1.0 | 19 | 116 | Behavioral, physiological, and neural signatures of surprise during naturalistic sports viewing | 10.1016/j.neuron.2020.10.029 |
| ds003340 | 10.18112/openneuro.ds003340.v1.0.2 | 18 | 142 | Tasting Pictures: Viewing Images of Foods Evokes Taste-Quality-Specific Activity in Gustatory Insular Cortex | 10.1073/pnas.2010932118 |
| ds003342 | 10.18112/openneuro.ds003342.v1.0.0 | 18 | 187 | Hand-selective visual regions represent how to grasp 3D tools for use: brain decoding during real actions | 10.1523/JNEUROSCI.0083-21.2021 |
| ds003521 | 10.18112/openneuro.ds003521.v1.0.0 | 35 | 35 | Watching Friday Night Lights (Study 2) | 10.1126/sciadv.abf7129 |
| ds003524 | 10.18112/openneuro.ds003524.v1.0.0 | 12 | 24 | Watching Friday Night Lights (Study 1) | 10.1126/sciadv.abf7129 |

## C.2   DATASETS FOR DOWNSTREAM TASK

A brief overview of the mental states included in both downstream datasets is provided below. This information is also cited from Thomas et al. (2022). For additional details on the experimental procedures of these datasets, readers are referred to the original publications: Van Essen et al. (2013) for HCP, King et al. (2019) for MDTB, and Tomoya Nakai (2020) for Over100.

**HCP.** Table 8 provides an overview of the mental states of each HCP tasks.

Table 8: HCP mental states. The mental states and total number of them are listed for each task.

| Task | Mental States | Count |
|---|---|---|
| Working memory | body, faces, places, tools | 4 |
| Gambling | win, loss | 2 |
| Motor | left / right finger, left / right toe, tongue | 5 |
| Language | story, math | 2 |
| Social | interaction, no interaction | 2 |
| Relational | relational, matching | 2 |
| Emotion | fear, neutral | 2 |

**MDTB.** The MDTB dataset comprises the following set of tasks, each representing a distinct mental state in our analyses (as labeled by the original authors): CPRO, GoNoGo, ToM, actionObservation, affective, arithmetic, checkerBoard, emotionProcess, emotional, intervalTiming, landscapeMovie, mentalRotation, motorImagery, motorSequence, nBack, nBackPic, natureMovie, prediction, rest, respAlt, romanceMovie, spatialMap, spatialNavigation, stroop, verbGeneration, and visualSearch.

**Over100.** In the Over100 dataset, tasks related to natural behaviors are designed as follows: PressRight, PressLeft, PressLR, RestOpen, RestClose, EyeBlink, RateTired, RateConfidence, RateSleepy, ImagineFuture, and 93 other tasks.

## C.3    DATASETS OF RESTING STATE

To conduct the analysis described in Section 4.2.1, we have constructed a new resting-state fMRI dataset. Following the methodology outlined by Thomas et al. (2022), we gathered data from OpenNeuro.org. The details of this dataset are presented in Table 9. The numbers of subjects and runs listed reflect those that were successfully preprocessed, excluding any that encountered errors during this stage.

In addition to these datasets, we utilized the "SRPBS Traveling Subject MRI Dataset," which comprises 411 scans of 3T MRI imaging data from 9 traveling subjects collected across 9 sites. For more details, refer to Tanaka et al. (2021). Note that due to redistribution restrictions, our shared dataset does not include this specific dataset. From this dataset, we obtained a total of 410 runs from the 9 subjects.

Data used in the preparation of this work were obtained from the DecNef Project Brain Data Repository (https://bicr-resource.atr.jp/srpbsts/) gathered by a consortium as part of the Japanese Strategic Research Program for the Promotion of Brain Science (SRPBS) supported by the Japanese Advanced Research and Development Programs for Medical Innovation (AMED)

## C.4    FMRI DATA PREPROCESSING

As previously mentioned, we utilized the datasets distributed by Thomas et al. (2022) for both upstream and downstream learning. However, for the resting state dataset, we prepared new data ourselves. In doing so, we adopted the same preprocessing methods as Thomas et al. (2022). The details are outlined below.

All fMRI data were preprocessed using fMRIPrep (versions 20.2.0 and 20.2.3; a minimal, automated preprocessing pipeline for fMRI data (Esteban et al. (2019))) with its default settings, excluding FreeSurfer (Fischl (2012)) surface preprocessing. Subsequently, we applied a series of additional minimal processing steps to the fMRIPrep derivatives, including i) spatial smoothing of the fMRI sequences with a 3mm full-width at half maximum Gaussian kernel, ii) detrending and high-pass filtering (at 0.008 Hz) of the individual voxel activity time courses, and iii) basic confound removal by regressing out noise related to head movement (as

indicated by the six basic motion regressors x, y, z, roll, pitch, and yaw) as well as the mean global signal and mean signals for white matter and cerebrospinal fluid masks (as estimated by fMRIPrep). Finally, each preprocessed fMRI run was parcellated using the DiFuMo atlas (see section 2.2 of the main text) and the resulting individual network time courses were standardized to have a mean of 0 and unit variance.

Table 9: Overview of resting state fMRI datasets. For each dataset, the OpenNeuro.org identifier and DOI are provided, along with the number of individuals and fMRI runs included in our upstream dataset, a brief text descriptor, and the DOI of an associated publication.

| ID | DOI | #Individuals | #Runs | Text descriptor | DOI publication |
|---|---|---|---|---|---|
| ds000221 | 10.18112/openneuro.ds000221.v1.0.0 | 61 | 215 | A mind-brain-body dataset of MRI, EEG, cognition, emotion, and peripheral physiology in young and old adults | 10.1038/sdata.2018.308 |
| ds000243 | null | 119 | 201 | Magnitudes and consequences for artifact detection | 10.1371/journal.pone.0182939 |
| ds001747 | 10.18112/openneuro.ds001747.v1.1.0 | 90 | 90 | Exploring the Resting State Neural Activity of Monolinguals and Late and Early Bilinguals | ISSN: 2572-4479 |
| ds003469 | 10.18112/openneuro.ds003469.v1.0.0 | 80 | 80 | Neurophysiological aspect in normal population | unpublished |
| ds003831 | 10.18112/openneuro.ds003831.v1.0.0 | 72 | 72 | Cognitive Control Theoretic Mechanisms of Real-time fMRI-Guided Neuromodulation (CTM) | NSF Award# BCS-1735820 |
| ds003974 | 10.18112/openneuro.ds003974.v3.0.0 | 52 | 52 | The Reading Brain Project L1 Adults | 10.1016/j.neuroling.2018.03.005 |
| ds003988 | 10.18112/openneuro.ds003988.v1.0.0 | 56 | 56 | The Reading Brain Project L2 Adults | 10.1016/j.neuroling.2018.03.005 |
| ds004349 | 10.18112/openneuro.ds004349.v1.0.0 | 54 | 54 | Evaluating methods for measuring background connectivity in slow event-related fMRI designs | 10.1002/brb3.3015 |

# D UPSTREAM LEARNING DETAILS

## D.1 TRAINING RESULTS

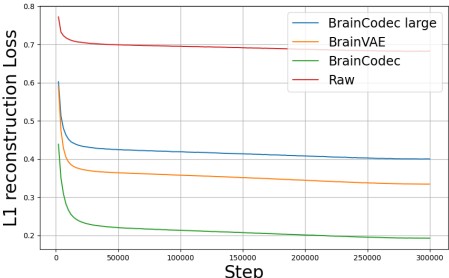 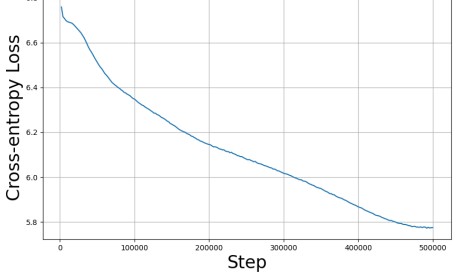

(a) The training loss in upstream learning of CSM.     (b) The training loss in upstream learning of TokenCSM.

Figure 11: Loss curves of the upstream learning of CSM & TokenCSM.

Here, we present the loss curves of upstream learning for the CSM models ultimately used in downstream learning.

Figure 11a shows the loss curves for the original CSM model, identical to the one used in the previous study by Thomas et al. (2022) (labeled as Raw in the figure), and for CSM combined with various codec models: CSM+BrainCodec, CSM+BrainCodec large, and CSM+BrainVAE. As shown in the figure, the Raw model exhibits the highest loss, while CSM+BrainCodec has the lowest. This result aligns with our motivation, demonstrating the beneficial effect of compression methods in facilitating model training. The Raw model is noisier and thus more difficult to predict accurately, leading to poorer generalization of CSM. CSM+BrainCodec large and CSM+BrainVAE, while able to reconstruct higher-resolution images due to lower compression ratios, result in increased loss during training.

Figure 11b presents the results for TokenCSM. Unlike CSM, TokenCSM operates on a token basis, preventing the use of BrainCodec large, which has a high number of codebooks, and BrainVAE, which is not a discretization method. Additionally, raw data cannot be tokenized, leaving us with a single result from tokenization using BrainCodec. While there are no comparisons here, the noteworthy aspect is the absolute value of the loss. The cross-entropy loss only decreases to around 6, which is quite large compared to other fields. For a detailed discussion on this, refer to B.3.

## D.2 GENERATED FMRI BY CSMS

As explained in Section 2.1, CSM is a model that predicts the current time-step data using past data. By using this model, we can input a portion of fMRI data and output the continuation. In this study, we used half of the fMRI data in an evaluation dataset as a prompt (condition) and generated the subsequent fMRI data.

Figure 12 shows examples of these generated sequences. When visualizing, the first two steps (the top two images in each figure) are used as the prompt. The subsequent images are generated data. Additionally, the ground truth fMRI data is displayed for comparison (left column).

First, compare Figures 12b and 12a. These figures show the results for CSM, which is the CSM trained using only raw fMRI data. As demonstrated by the high loss shown in Section D.1, the generated data significantly deviates from the ground truth.

Next, compare Figures 12d and 12c. These show the results for CSM+BrainCodec. Note that the ground truth data (Figure 12d) is already reconstructed data. At the initial step following the prompt, the generated data closely matches the ground truth. However, as the sequence progresses, the generated data increasingly diverges from the ground truth. This divergence is likely due to the autoregressive nature of the CSM, where errors accumulate during sequential generation. Alternatively, this divergence could represent a plausible progression of fMRI data, but this was not verified in this study.

Finally, compare Figures 12f and 12e. These figures show the results for CSM+BrainVAE. The generated results are more detailed than those using BrainCodec due to the use of BrainVAE. However, discrepancies from the ground truth are noticeable even from the step immediately following the prompt.

In conclusion, given the current data volume and model configurations, using BrainCodec is likely to provide a level of learning capable of generating plausible sequences. Of course, this conclusion cannot be drawn from a single example. For instance, generating data for downstream learning and performing classification on this generated data could objectively compare the quality of the generated sequences. This remains a subject for future research.

# E DOWNSTREAM LEARNING DETAILS

## E.1 ALL RESULTS WITH LINEAR MODEL

As mentioned in Section 4.1.2, the performance of the Linear model was highly influenced by weight decay, likely due to the model's small size. In Section 4.1, we could only include part of the results due to space constraints. Here, we present the complete results.

First, consider the results for the HCP dataset shown in Table 10. It is important to note that as we selected the best method for each approach to draw the previous conclusions, the conclusions remain unchanged here. One noticeable point we can find from this table is the clear advantage of the compression methods. For instance, in the Linear model without any codec, the best result is achieved with a weight decay of 1. This is natural since the model is very small, the data is noisy, and some weight decay helps prevent overfitting by acting as a regularization term. On the other hand, the Linear+BrainCodec model achieves the best performance with the

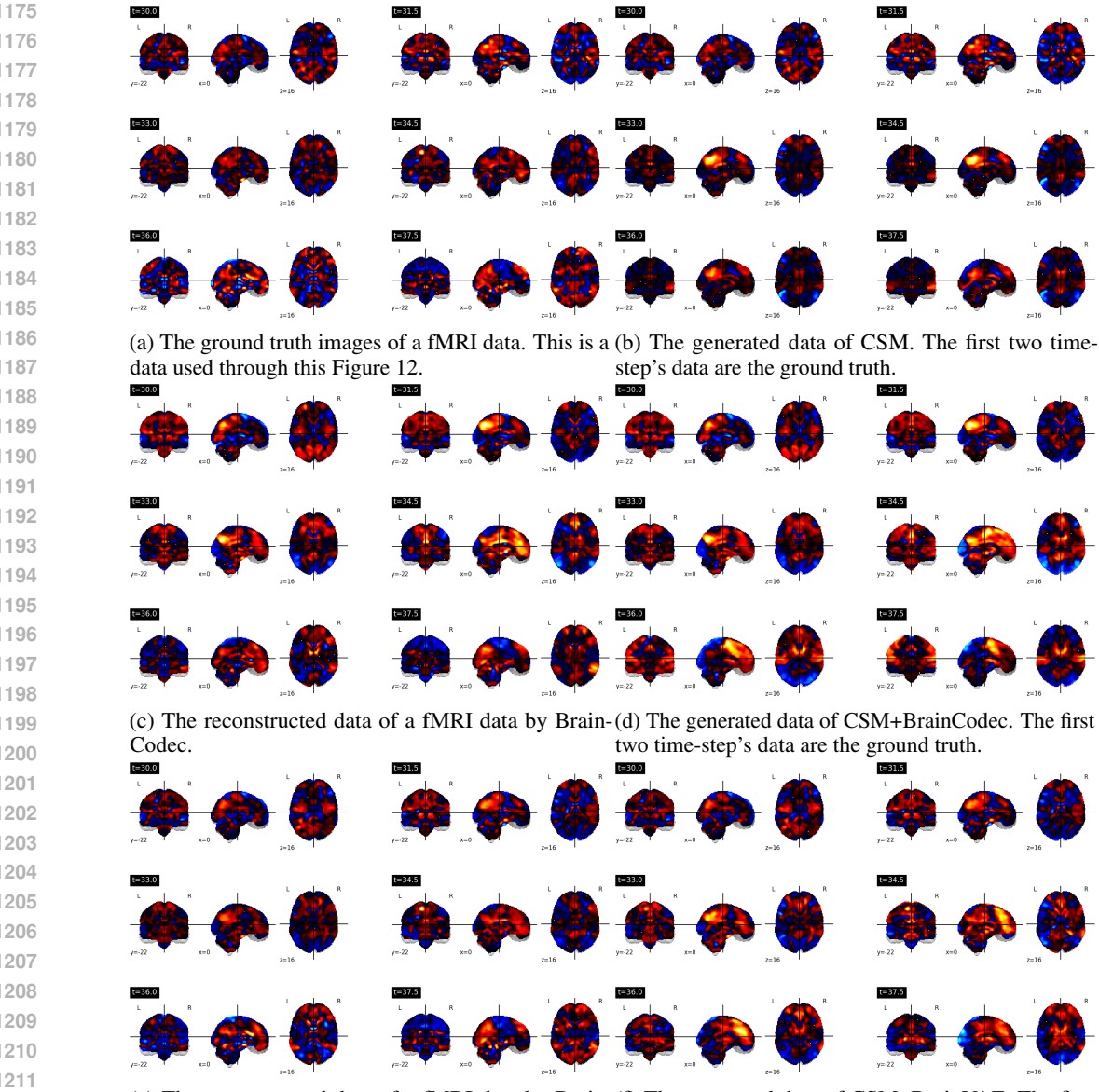

(a) The ground truth images of a fMRI data. This is a data used through this Figure 12.

(b) The generated data of CSM. The first two time-step's data are the ground truth.

(c) The reconstructed data of a fMRI data by Brain-Codec.

(d) The generated data of CSM+BrainCodec. The first two time-step's data are the ground truth.

(e) The reconstructed data of a fMRI data by Brain-VAE.

(f) The generated data of CSM+BrainVAE. The first two time-step's data are the ground truth.

Figure 12: The generated fMRI data by CSM. These are the continuation from the ground truth.

smallest weight decay. This is because BrainCodec compresses and reconstructs the data, retaining important information and making it easier for the model to learn. Hence, in the HCP dataset, the codec method proves to be a very powerful tool. We can see the same trend with MDTB dataset.

In contrast, the results for the Over100 dataset shown in Table 10 are somewhat different. For example, while the smallest weight decay yields the best results in the baseline, it is not the best for BrainCodec, which

Table 10: Decoding performances when combining a linear model with various codec models and changing its weight decay.

| Codec | weight decay | HCP | | MDTB | | Over100 | |
|---|---|---|---|---|---|---|---|
| | | Acc | F1 | Acc | F1 | Acc | F1 |
| - | 10 | 0.549 | $0.510 \pm 0.165$ | 0.793 | $0.791 \pm 0.076$ | 0.146 | $0.012 \pm 0.045$ |
| - | 1 | **0.622** | **$0.616 \pm 0.161$** | **0.825** | **$0.827 \pm 0.069$** | 0.256 | $0.155 \pm 0.140$ |
| - | 0.1 | 0.582 | $0.593 \pm 0.170$ | 0.811 | $0.812 \pm 0.072$ | **0.259** | **$0.156 \pm 0.136$** |
| BrainVAE | 10 | 0.650 | $0.604 \pm 0.220$ | 0.793 | $0.792 \pm 0.074$ | 0.146 | $0.012 \pm 0.043$ |
| BrainVAE | 1 | **0.734** | **$0.721 \pm 0.165$** | **0.823** | **$0.825 \pm 0.072$** | 0.280 | **$0.179 \pm 0.145$** |
| BrainVAE | 0.1 | 0.701 | $0.698 \pm 0.177$ | 0.817 | $0.818 \pm 0.072$ | **0.282** | $0.177 \pm 0.148$ |
| BrainCodec large | 10 | 0.636 | $0.580 \pm 0.244$ | 0.791 | $0.790 \pm 0.076$ | 0.147 | $0.013 \pm 0.044$ |
| BrainCodec large | 1 | **0.750** | **$0.735 \pm 0.168$** | **0.828** | **$0.829 \pm 0.069$** | **0.317** | **$0.206 \pm 0.154$** |
| BrainCodec large | 0.1 | 0.722 | $0.717 \pm 0.170$ | 0.816 | $0.818 \pm 0.071$ | 0.306 | $0.195 \pm 0.149$ |
| BrainCodec | 10 | 0.570 | $0.502 \pm 0.189$ | 0.793 | $0.791 \pm 0.076$ | 0.145 | $0.011 \pm 0.039$ |
| BrainCodec | 1 | 0.770 | $0.744 \pm 0.156$ | 0.803 | $0.802 \pm 0.069$ | **0.319** | **$0.213 \pm 0.153$** |
| BrainCodec | 0.1 | **0.814** | **$0.784 \pm 0.133$** | **0.811** | **$0.812 \pm 0.072$** | 0.302 | $0.193 \pm 0.150$ |

contrasts with the trends observed in HCP and MDTB. This is likely due to the higher difficulty of the dataset. Since Over100 involves classifying over 100 classes, it is much more challenging than other datasets, and even with BrainCodec, there is a risk of overfitting. Therefore, the smallest weight decay is not optimal. In contrast, in the baseline models, there is little difference between weight decay values of 1 and 0.1, which suggests that the difficulty of the problem is so high that overfitting is not yet an issue.

Thus, neuroscience datasets exhibit significantly different characteristics, and improved generalization performance can be expected through the collection of even larger datasets in the future.

## F    CODEBOOK ANALYSIS DETAILS

### F.1    ALL RESULTS OF UMAP VISUALIZATION OF CODEBOOKS

In Section 4.2.1, we used UMAP to visualize the codebooks learned by BrainCodec from resting state and task-related brain activities. This was done to verify the neuroscientific knowledge that brain activity during resting state encompasses task-related brain activity. We noted that similar patterns were observed in the second and subsequent codebooks. Here, we present the complete visualization of all codebooks in Figure 13. As expected, the overall conclusions remain unchanged.

### F.2    OBJECTIVE EVALUATION OF REST-CODEBOOK & TASK-CODEBOOK WITH SINKHORN ALGORITHM

To quantify the similarity between codebooks, we employed the Sinkhorn distance (Cuturi (2013)), calculated using an L2 distance cost matrix that underwent Min-Max normalization, allowing codebook-wise absolute comparisons. The Sinkhorn algorithm was then applied to compute the Sinkhorn distance.

The results, displayed in Fig 14, indicate that when comparing Task-codebook and Rest-codebook, the overall Sinkhorn distance is smaller than a reference (distances between Random-codebooks and distances to the Random-codebook). RVQ inherently clusters low-frequency components, meaning that major variations typically aggregate in the lower codebooks (A.1). The close distances between the first and second codebooks may suggest that the primary variations in resting state fMRI and task fMRI are relatively similar. Additionally,

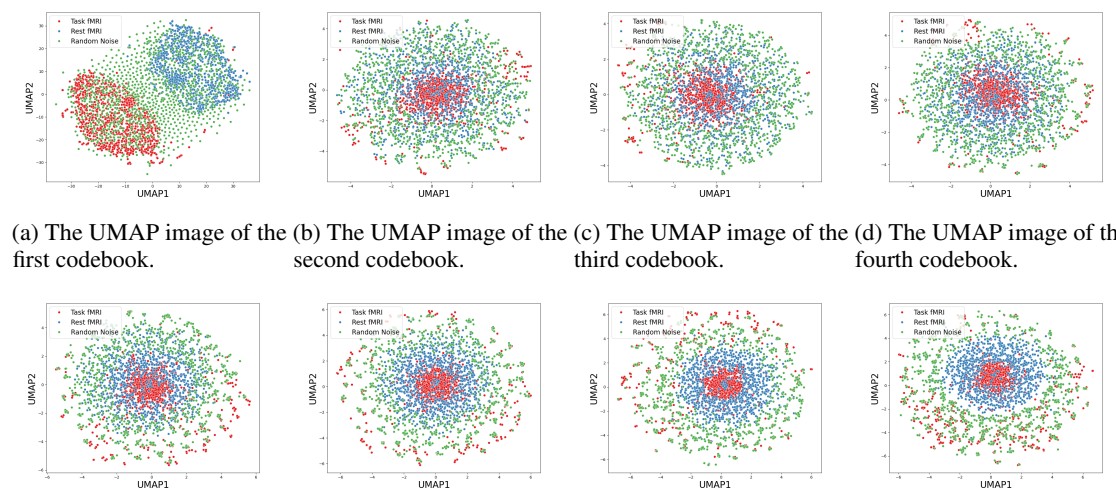

(a) The UMAP image of the first codebook. (b) The UMAP image of the second codebook. (c) The UMAP image of the third codebook. (d) The UMAP image of the fourth codebook.

(e) The UMAP image of the fifth codebook. (f) The UMAP image of the sixth codebook. (g) The UMAP image of the seventh codebook. (h) The UMAP image of the eighth codebook.

Figure 13: The UMAP visualization of all codebooks compares Task-codebook (red), Rest-codebook (blue), and Random-codebook (green).

the fact that higher codebooks have distances closer to the Random-codebook distances indicates the separation capability of RVQ.

However, since the comparison is made with Gaussian noise, the smaller distance does not necessarily conclude that resting state fMRI and task fMRI are similar. More appropriate comparisons are needed, which is a subject for future research.

### F.3 THE OTHER EXAMPLES OF RECONSTRUCTED FMRI BY PARTIAL CODEBOOKS OF BRAINCODEC

In Section 4.2.2, we evaluated BrainCodec's reconstruction capability by illustrating reconstructed images for a specific HCP task fMRI. We demonstrated that these reconstructions make brain activity more discernible compared to the original or VAE-based methods. Here, we will show that the previous example is not an isolated success by presenting more cases using HCP data from different tasks.

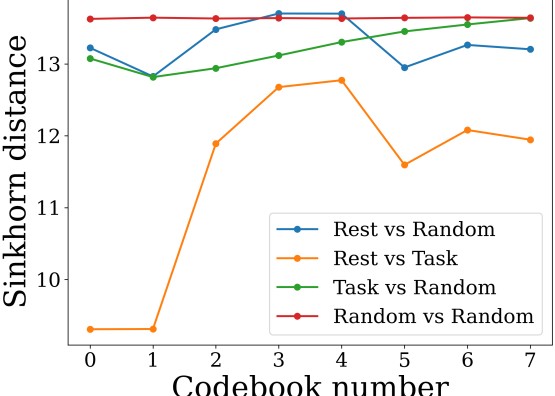

Figure 14: The Sinkhorn distance between Task-codebook and Rest-codebook. For comparison, distances between Random-codebooks and distances to the Random-codebook are also displayed.

For consistency, we set the brain slicing positions to align with the results published in Barch et al. (2013), which analyzes HCP task fMRI data. For example, the previous MOTOR task is depicted in Fig. 7 of this paper. In this paper, brain activity is visualized using traditional methods that average data over time and different runs, allowing for detailed identification of activity locations in the motor cortex. However, this traditional approach does not reveal the progression of brain activity starting from the visual cortex, as seen in Section 4.2.2. Therefore, when reviewing the results of the reconstructed images, please refer to both the

traditional method as cited in Barch et al. (2013) and the averaged fMRI data, which, despite being a simple average, takes into account temporal variations. In the following sections, we describe the results respecting the traditional methods; however, it should be noted here that there is consistency between the results of the traditional methods and our averaged fMRI data throughout.

Below, we present reconstructed images for other tasks: Language, Social, and Relational tasks.

Fig 15 presents the reconstruction results for the Language task. The Language task is divided into "story" and "math" components, and here we show the results for the "story" component. In the "story" task, participants first listen to a short story adapted from Aesop's fables. Following this, they answer multiple-choice questions related to the story. For more detailed information about this task, please refer to Binder et al. (2011).

Upon examining the figure, it is evident that the reconstructions by BrainVAE closely resemble the originals, failing to effectively suppress noise. In contrast, the reconstructions by BrainCodec reveal strong reactions in the lower temporal regions that are not visible in the originals or BrainVAE reconstructions. These brain activities, observable only with BrainCodec, are crucial for characterizing the "story" task, as seen in Fig 8 of Barch et al. (2013). Additionally, using only the first half of the codebooks exhibits similar trends, suggesting that, as discussed in Section 4.2.2, this approach may be sufficient. As in Section 4.2.2, we report the L1 distance between the Mean and each fMRI data (similarly for other tasks introduced below). The order is the same: original, BrainVAE, BrainCodec (0-7), and BrainCodec (0-3), with the results as follows: $0.794 \pm 0.059, 0.824 \pm 0.042, 0.380 \pm 0.047, 0.362 \pm 0.038$. Indeed, similar to the MOTOR task, it can be confirmed that the reconstructions by BrainCodec replicate the Mean more accurately.

Next, Fig 16 shows the results for the Social task. In this task, participants watch a video displaying several shapes moving randomly on a screen. They are then asked to determine whether the shapes appear to be engaging in social interactions ("mental") or moving randomly without such interactions ("rnd"). For this reconstruction, we used the "mental" examples, where the shapes appear to be interacting. For further details on this task, please refer to Wheatley et al. (2007).

In this example, as well, the difference between the original and BrainVAE is minimal. In fact, BrainVAE appears to weaken the signal strength, suggesting that this might be a challenging example for VAE methods. On the other hand, the reconstructions by BrainCodec clearly identify areas of strong signal. Notably, in this example, there is widespread activation throughout the brain. While this might initially seem like excessive activation, it is consistent with the results shown in Fig 9 of Barch et al. (2013). Similarly, the same patterns are observed when using only the first half of the codebooks. Similarly, the L1 distances between the Mean and each fMRI data are as follows: $0.767 \pm 0.038, 0.841 \pm 0.028, 0.358 \pm 0.037, 0.343 \pm 0.032$. This further confirms the performance of BrainCodec numerically.

Finally, Fig 17 shows the results for the relational task. In this task, six types of shapes and six types of textures are used as stimuli. In the relational processing condition ("relation"), two pairs of objects are presented at the top and bottom of the screen. Participants first determine the dimension (shape or texture) in which the upper pair differs, and then decide whether the lower pair differs in the same dimension. In the control condition ("match"), two objects are shown at the top, one object at the bottom, and a word ("shape" or "texture") in the center. Participants must determine whether the bottom object matches either of the top objects in the specified dimension. For this reconstruction, we used data from the "relation" condition. For a more detailed explanation, please refer to Smith et al. (2007).

In this example, BrainVAE allows for relatively clear observation of brain activity, such as the responses in the visual cortex and the lower occipital regions (t=3.60). However, in all cases, BrainCodec exhibits these activities more distinctly. Notably, brain activity near the parietal region, reported in Fig 10 of Barch et al. (2013), is not observable in the original or BrainVAE reconstructions. This activity, however, is clearly visible in the BrainCodec reconstruction (t=5.76). Similarly, the L1 distances between the Mean and each fMRI

data are as follows: $0.765 \pm 0.038, 0.909 \pm 0.025, 0.353 \pm 0.033, 0.334 \pm 0.028$. The same results were confirmed.

The above are additional concrete examples demonstrating the superiority of our BrainCodec. These examples were chosen for their clarity. The overall effectiveness of this method, which can be viewed as a denoising technique, should ideally be evaluated quantitatively—a task for future research. Our code and trained BrainCodec models are publicly available, and we encourage you to apply them to your own fMRI data to assess their denoising performance.

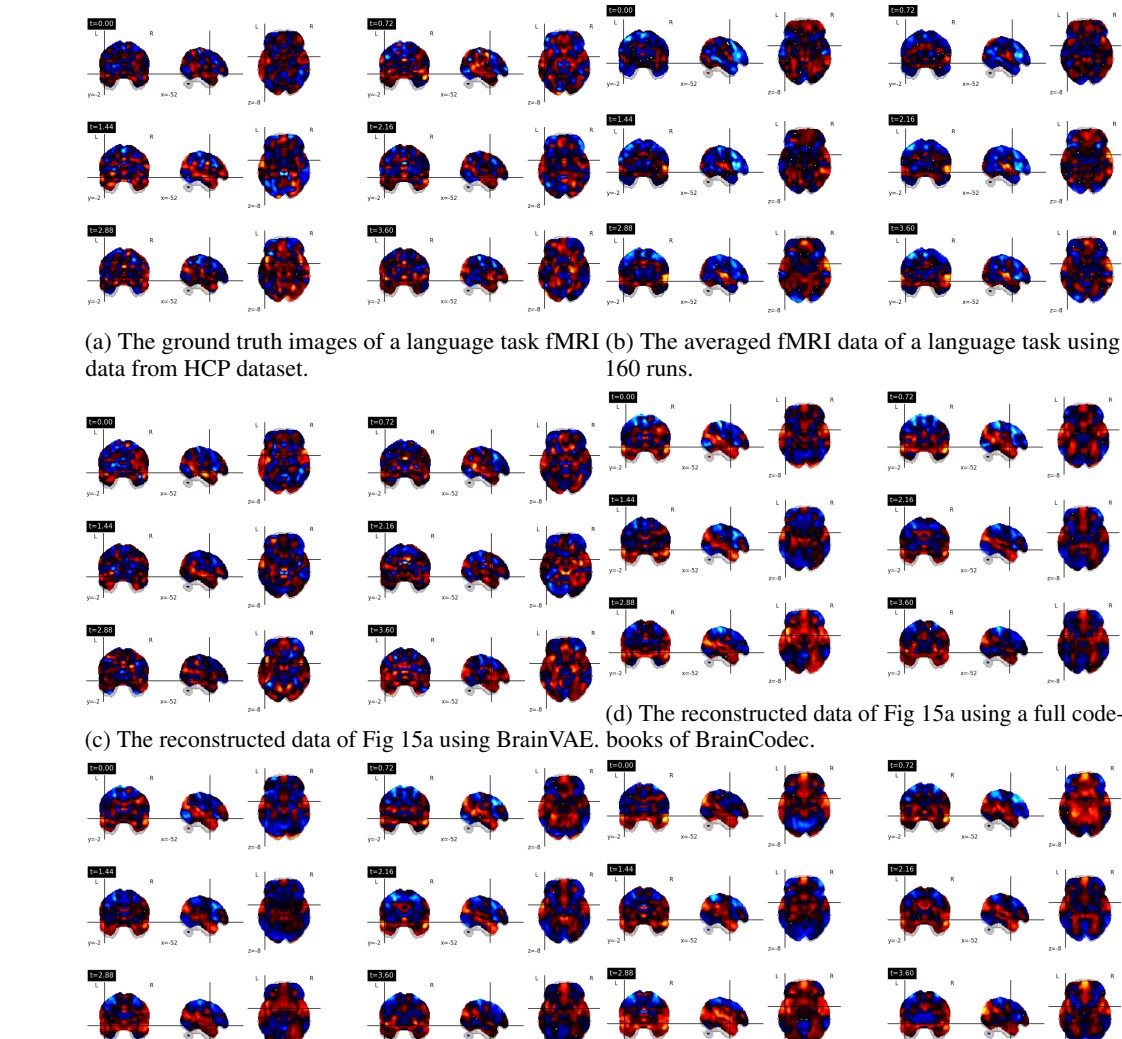

(a) The ground truth images of a language task fMRI data from HCP dataset.

(b) The averaged fMRI data of a language task using 160 runs.

(c) The reconstructed data of Fig 15a using BrainVAE.

(d) The reconstructed data of Fig 15a using a full code-books of BrainCodec.

(e) The reconstructed data of Fig 15a using a first half codebooks of BrainCodec.

(f) The reconstructed data of Fig 15a using a first code-book of BrainCodec.

Figure 15: Comparison of the ground truth fMRI data, the averaged fMRI data using 160 runs, and the reconstructed images using BrainVAE and BrainCodec for language task from the HCP dataset.

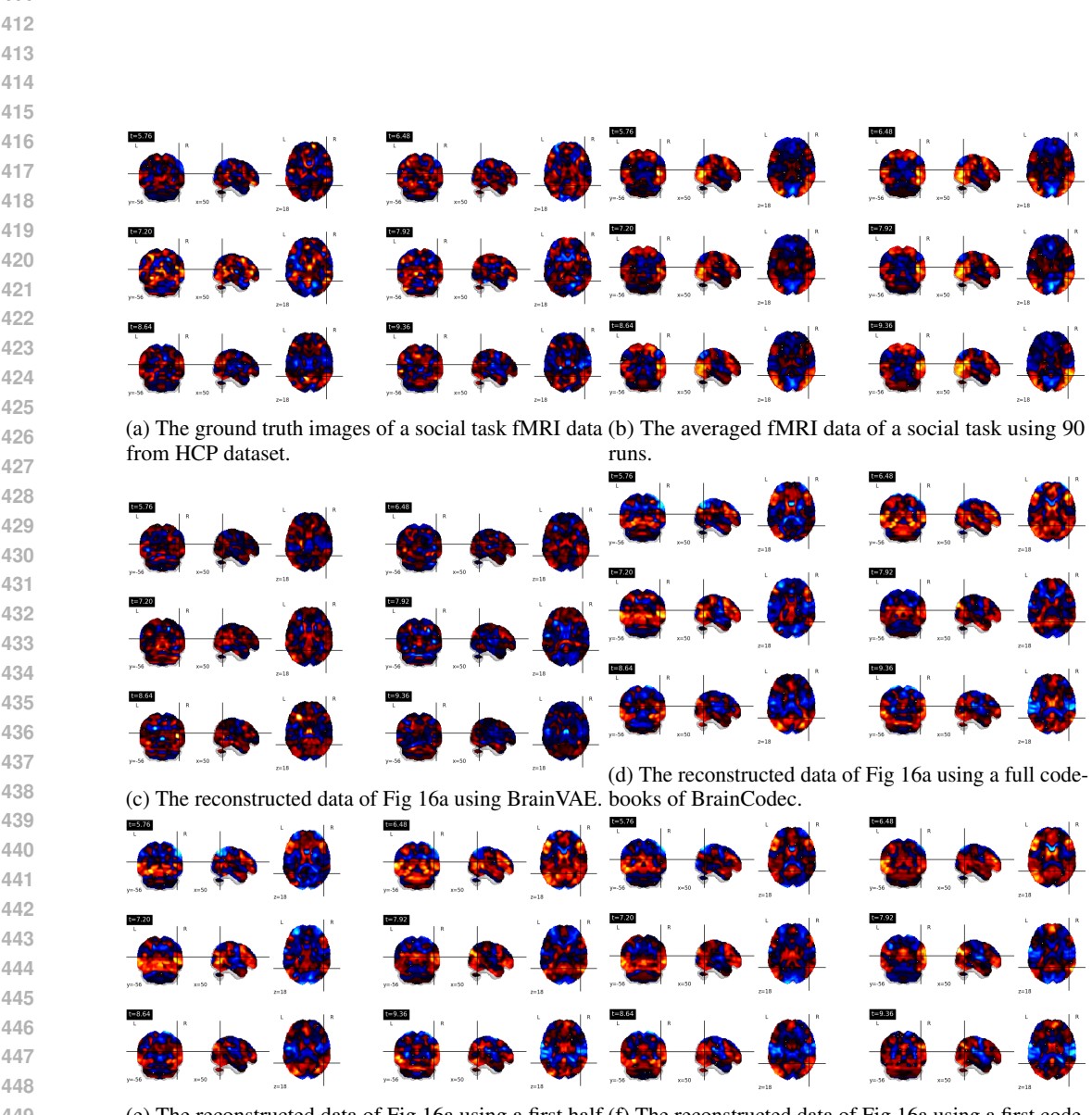

(a) The ground truth images of a social task fMRI data from HCP dataset.

(b) The averaged fMRI data of a social task using 90 runs.

(c) The reconstructed data of Fig 16a using BrainVAE.

(d) The reconstructed data of Fig 16a using a full codebooks of BrainCodec.

(e) The reconstructed data of Fig 16a using a first half codebooks of BrainCodec.

(f) The reconstructed data of Fig 16a using a first codebook of BrainCodec.

Figure 16: Comparison of the ground truth fMRI data, the averaged fMRI data using 90 runs, and the reconstructed images using BrainVAE and BrainCodec for social task from the HCP dataset.

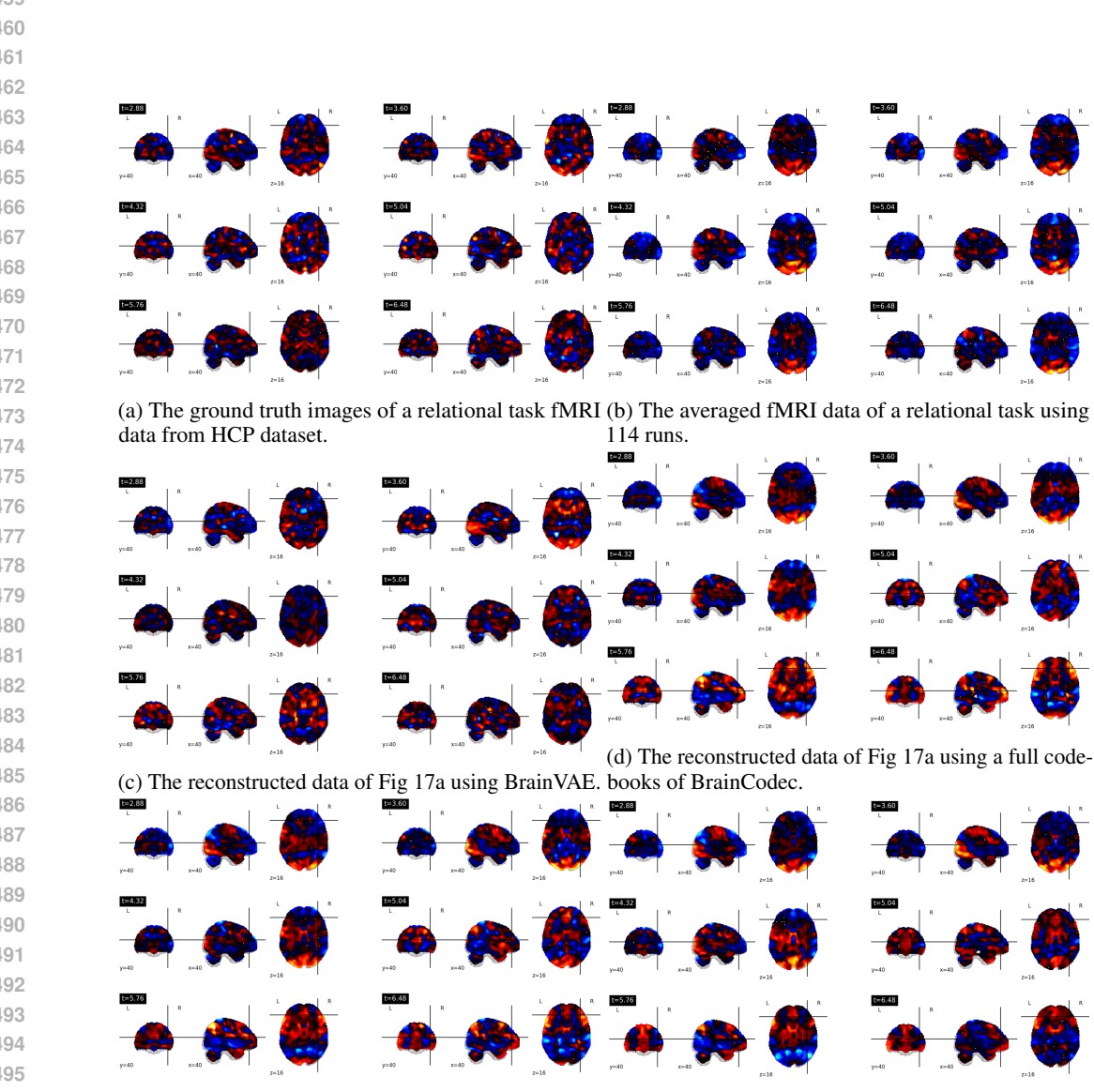

(a) The ground truth images of a relational task fMRI data from HCP dataset.

(b) The averaged fMRI data of a relational task using 114 runs.

(c) The reconstructed data of Fig 17a using BrainVAE.

(d) The reconstructed data of Fig 17a using a full codebooks of BrainCodec.

(e) The reconstructed data of Fig 17a using a first half codebooks of BrainCodec.

(f) The reconstructed data of Fig 17a using a first codebook of BrainCodec.

Figure 17: Comparison of the ground truth fMRI data, the averaged fMRI data using 114 runs, and the reconstructed images using BrainVAE and BrainCodec for relational task from the HCP dataset.