# OpenReview forum: "BrainCodec: Neural fMRI codec for the decoding of cognitive brain states"
_ICLR.cc/2025/Conference — ICLR 2025 Conference Withdrawn Submission_

### Official Review · Reviewer_CVdS · 2024-11-01

**Soundness:** 3
**Presentation:** 2
**Contribution:** 2
**Rating:** 3
**Confidence:** 5

**Summary:**

This work demonstrates a solid motivation to develop an innovative pipeline, BrainCodec, built on the Autodendoer (AOE) architecture to decode/reconstruct fMRI signals and provide neuroscientists with deeper insights. However, the reviewers have raised some key concerns about BrainCodec.

**Strengths:**

1. The proposed BrainCodec is novel, integrating some prevalent techniques such as  Residual Vector Quantization and Causal Sequence Modeling.

2. Based on reported empirical experiments, BrainCodeC can reconstruct fMRI signals better than BrainVAE.

**Weaknesses:**

The reviewers have highlighted several weaknesses in this work, which can be categorized as follows:

1). Missing Details: This concern encompasses Questions 1, 5, and 6, where critical information appears to be lacking.

2). Poor Motivation: This issue pertains to Question 2, where the rationale behind the codebook is not thoroughly explained. Additionally, reconstructing fMRI data does not provide comparable insights to those offered by fMRI analytics and diagnostics.

3). Technological Issues: Questions 3, 4, and 5 raise concerns about specific technological aspects, including optimization and hyperparameter tuning.

**Questions:**

The following 6 questions correspond to Weaknesses.

1). Individual Application: Since fMRI provides valuable insights into functional impairments faster than structural imaging, an effective pipeline for individual fMRI signal analysis and decoding would be highly beneficial. The authors appear to have developed a novel framework similar to AOE, which typically requires a large dataset for training, even in unsupervised learning settings. However, the authors do not discuss the applicability of their framework to individual fMRI data. If it cannot be applied on a personal level, the framework’s clinical translational potential may be limited.

2). The Concept of Codebooks: The authors applied their framework to both resting-state and task-based fMRI to establish a codebook explaining neural activity. However, they provide only qualitative explanations and visualizations. Reviewers suggest that the authors could strengthen the connection between the codebook and established fMRI components, such as Brain Connectivity Networks (BCNs) or specific task paradigms. The reviewers would like to see the codebook serve as an alternative technique for representing brain activity. For example, task-codebooks could be compared with task paradigms and be linked with spatial features, like BCNs, to clarify their role in fMRI analytics. Importantly, reviewers are particularly interested in understanding whether codebooks could contribute to early diagnostics for neurological disorders, such as Alzheimer’s Disease (AD) and Traumatic Brain Injury (TBI). Meanwhile, the codebook concept should be introduced and contextualized in the introduction section.

3). Hyperparameter Tuning: It appears that essential hyperparameters in this work were determined manually. Reviewers are concerned that some hyperparameters, such as the number of artificial neurons, could significantly impact the Braincodes's outcomes. A detailed description of the hyperparameter tuning process would enhance transparency and reproducibility.

4). Optimization: The authors presented a complex model but did not provide sufficient details about the optimization approach used in this work. Reviewers would like clarification on the choice of advanced gradient-based optimizers (e.g., Adam, STORM, or SGD), along with specifics on the learning rate and other vital parameters.

5). Datasets: The authors did not include detailed demographic information on the datasets used in their experiments. Reviewers recommend providing specifics on the number of subjects, as well as age and gender distributions.

6). Empirical Experimentation: Reviewers found the experimental design somewhat confusing. In section 4.2.1, the authors report codebooks from resting-state fMRI, yet multiple task-based codebooks are mentioned within this section. Reviewers suggest that results from resting-state and task-based fMRI should be reported separately, as these data types are distinct. Additionally, the authors only compare the reconstructed fMRI data using BrainCodec with BrainVAE. Including comparisons with other representative linear learning methods, such as Independent Component Analysis (ICA) and Sparse Dictionary Learning (SDL), would allow for a more comprehensive assessment of BrainCodec’s time efficiency and reconstruction accuracy.

---

### Official Review · Reviewer_E6wA · 2024-11-02

**Soundness:** 2
**Presentation:** 3
**Contribution:** 2
**Rating:** 5
**Confidence:** 4

**Summary:**

The paper introduces BrainCodec, a novel compression technique for functional magnetic resonance imaging (fMRI) data, aimed at enhancing mental state decoding. The authors address the challenges posed by small fMRI datasets and low signal-to-noise ratios (SNR) by proposing BrainCodec as a preprocessing step. They demonstrate that BrainCodec improves mental state decoding performance and provides valuable insights into latent representations, differentiating between task and resting state fMRI.

**Strengths:**

The paper has some strengths, such as the innovative introduction of BrainCodec as a compression technique for fMRI data, which addresses important challenges like small dataset sizes and low signal-to-noise ratios (SNR). The approach shows promise in improving mental state decoding performance and offers insights into latent representations, highlighting differences between task and resting state fMRI. Additionally, the suggestion that BrainCodec could function as a denoising method indicates potential for further research.

**Weaknesses:**

The paper presents several limitations that undermine its overall contribution. Firstly, it fails to clearly distinguish the contributions of BrainCodec from existing methods, particularly those discussed in Thomas et al. (2022), including Causal State Modeling (CSM) and linear approaches. This lack of differentiation raises questions about the novelty and significance of the proposed method. Secondly, the evaluation lacks a thorough comparison with established techniques in fMRI analysis, making it difficult to validate the effectiveness of BrainCodec and to position it within the current landscape of fMRI processing methodologies. Lastly, the paper does not provide sufficient details about the experimental setup, particularly regarding the processing and parcellation of fMRI data into regions of interest (ROIs), which is crucial for reproducibility and for understanding the implications of the findings.

**Questions:**

1. Limited Novelty and Contribution: The paper fails to clearly distinguish the contributions of BrainCodec from existing methods, particularly those discussed in Thomas et al. (2022), which include linear approaches and Causal State Modeling (CSM). The authors should articulate the specific advancements that BrainCodec offers in relation to prior work to establish its significance.

2. Inadequate Comparison with Established Techniques: The evaluation lacks a thorough comparison with state-of-the-art techniques in fMRI analysis. A robust comparative analysis is essential to validate the proposed method’s effectiveness and situate it within the current landscape of fMRI processing methodologies.

3. Insufficient Methodological Details: The paper does not provide adequate details about the experimental setup, especially regarding the processing and parcellation of fMRI data into regions of interest (ROIs). A clearer methodology is crucial for reproducibility and understanding the implications of the findings.

---

### Official Review · Reviewer_VZm2 · 2024-11-04

**Soundness:** 2
**Presentation:** 3
**Contribution:** 3
**Rating:** 5
**Confidence:** 3

**Summary:**

This study introduces BrainCodec, a novel fMRI codec inspired by neural audio codecs, aimed at improving mental state decoding from fMRI data. Given the small scale of fMRI datasets and their low signal-to-noise ratios (SNR), the researchers apply compression techniques as a preprocessing step. The evaluation of BrainCodec shows significant improvements in decoding performance compared to previous methods. Additionally, the analysis of latent representations reveals insights into the similarities and differences between task and resting state fMRI, enhancing interpretability. The study also demonstrates that BrainCodec can improve SNR, suggesting its potential as an effective denoising method and offering new analytical opportunities in neuroscience.

**Strengths:**

- The paper is generally well-organized and easy to follow.
- The motivation and objectives of the study are clearly articulated and compelling.
- The compression approach proposed in this study appears promising.

**Weaknesses:**

- While the authors compare BrainCodec with BrainVAE and the baseline methods, a more comprehensive comparison with other fMRI compression techniques, such as dictionary learning-based methods, could provide a deeper understanding of the relative strengths and weaknesses.
- The authors do not fully address the limitations of the current study, unless I missed something.

**Questions:**

- The performance of BrainCodec seems to vary across different datasets, with the MDTB dataset showing less improvement compared to the HCP and Over100 datasets. The authors acknowledge the dataset-specific characteristics, but further investigation or more discussion into the factors affecting performance would be beneficial.
- For Table4, besides of L1 dist, spatial correlation could also be a potential metric from a different aspect to evaluate the similarity between two patterns, which might provide a more comprehensive evaluation.
- What would happen if the codebook number continue increasing, like to 12, 16, etc.

---

### Official Review · Reviewer_zQKp · 2024-11-09

**Soundness:** 3
**Presentation:** 2
**Contribution:** 3
**Rating:** 5
**Confidence:** 3

**Summary:**

The paper introduces BrainCodec, a novel fMRI data compression technique inspired by neural audio codecs. The authors demonstrate improved performance on mental state decoding tasks when combining BrainCodec with existing models like linear classifiers and Causal Sequence Models. They also analyze the learned codebooks to show interpretable representations of brain activity patterns.

**Strengths:**

- Novel application of audio compression techniques to fMRI data
- Building up on one of the largest "foundation models"
- Improved performance on mental state decoding tasks when combined with existing models
- Comprehensive ablation studies and hyperparameter analysis
- Code and trained models made publicly available

**Weaknesses:**

- Lack of comparison to relevant baseline methods like temporal ICA or sparse dictionary approaches
- Inconsistent performance improvements across all datasets (e.g., MDTB showed decreased performance)
- Potential overfitting concerns when using small models like Linear+BrainCodec
- The paper's organization, clarity, or depth of discussion may not have met the high standards expected for the conference, for example "Our contributions" section as presented does not effectively communicate novel contributions, but rather reads like a summary of the paper's contents.

Missing key (very relevant) literature
 - (Encodec mentioned but not cited) High Fidelity Neural Audio Compression (https://arxiv.org/abs/2210.13438)
 - Neural Codec Language Models are Zero-Shot Text to Speech Synthesizers (https://arxiv.org/pdf/2407.06244)

**Questions:**

How does BrainCodec compare to other fMRI compression or denoising techniques like temporal ICA or sparse dictionary approaches?

How sensitive is the method to the choice of hyperparameters, particularly the number of codebooks?

What are the computational requirements and processing time for BrainCodec?

Different cognitive tasks have different durations/stimuli blocks. Would sampling all tasks to 50 timepoints disproportionately affect certain task types?

Could the matching of TRs to a predetermined range lead to slight misalignments between the actual timing of fMRI data and its representation in the model? I am having hard time wrapping my head around this.

---

### Note · Authors · 2024-11-27

I have read and agree with the venue's withdrawal policy on behalf of myself and my co-authors.